# Genomic insights into the taxonomic status and bioactive gene cluster profiling of *Bacillus velezensis* RVMD2 isolated from desert rock varnish in Ma'an, Jordan

Sulaiman M. Alnaimat[1], Saqr Abushattal[1], Saif M. Dmour[1], Wajdy J. Al-Awaida[2]*, Amani M. Ayyash[3], Khang Wen Goh[4]

1 Department of Medical Analysis, Princess Aisha Bint Al-Hussein College of Nursing and Health Sciences, Al-Hussein Bin Talal University, Ma'an, Jordan, 2 Department of Biology and Biotechnology, Faculty of Science, American University of Madaba, Madaba, Jordan, 3 Department of Pharmacy, Faculty of Health Sciences, American University of Madaba, Madaba, Jordan, 4 Faculty of Data Science and Information Technology, INTI International University, Nilai, Malaysia

* w.alawaida@aum.edu.jo

## Abstract

Extreme environments like arid and semi-arid deserts harbor unique microbial diversity, offering rich sources of specialized microbial metabolites. This study explores *Bacillus velezensis* RVMD2, a strain isolated from rock varnish in the Ma'an Desert, Jordan. The genome was sequenced using the Illumina NextSeq 2000 platform, resulting in a 4,212,579 bp assembly with a GC content of 45.94%. The assembled genome comprises 112 contigs and encodes 4,250 proteins, 77 tRNA genes, and 4 rRNA genes. Phylogenetic analysis of the 16S rRNA gene indicated a 99.84% similarity to previously identified *B. velezensis* strains. Whole-genome phylogeny using EzBiome, MiGA, and TYGS confirmed its classification as *B. velezensis*. Functional annotation identified genes involved in carbohydrate metabolism, including 324 carbohydrate-active enzyme (CAZyme) genes, stress response, and secondary metabolite biosynthesis. The genome also contains 50 genes associated with heavy metal resistance and plant growth promotion. Analysis using AntiSMASH identified 12 biosynthetic gene clusters involved in the production of secondary metabolites, including fengycin, surfactin, polyketides, terpenes, and bacteriocins. Notably, several clusters did not match any known sequences, suggesting the presence of potentially novel antimicrobial compounds. The genomic features of RVMD2 highlight its adaptability to extreme environments and its potential for biotechnological applications, including bioremediation and the discovery of novel bioactive metabolites.

**Data availability statement:** The genomic data for Bacillus velezensis strain RVMD2 were submitted to GenBank under the BioProject accession number PRJNA987486. The specific BioSample associated with this project is SAMN36891939. This Whole Genome Shotgun project has been deposited at DDBJ/ENA/GenBank under the accession JAVBHW000000000. The version described in this paper is version JAVBHW010000000. The raw fastq files from genome sequencing were deposited in the NCBI Sequence Read Archive (SRA) under accession number SRR25608412. Additionally, the 16S rRNA gene sequence is accessible under the GenBank accession number PP942176.

**Funding:** This research was funded by the Deanship of Scientific Research at Al-Hussein Bin Talal University, Jordan, under grant number 2022/119. The funders had no role in study design, data collection and analysis, decision to publish, or preparation of the manuscript.

**Competing interests:** The authors have declared that no competing interests exist.

## Introduction

Research interest in extreme environments, particularly arid and semi-arid deserts, has intensified due to their distinct microbial communities [1]. One of the remarkable features of these regions is rock varnish, a thin dark layer on rocks, typically less than 200 μm thick. This varnish is primarily composed of oxygen, silicon, and aluminum [2]. In recent years, studies have highlighted that microorganisms inhabiting such extreme environments can serve as significant reservoirs for specialized metabolites [3]. Genome mining of desert-isolated strains has enabled the identification of biosynthetic gene clusters (BGCs) linked to the production of diverse bioactive compounds, including antibiotics [4]. These findings underscore the potential of extreme environment-derived microorganisms as promising sources for novel antibiotic discovery.

The *Bacillus* genus includes Gram-positive bacteria that form spores and are well-known for producing bioactive substances, as well as promoting plant growth. Their antimicrobial properties are largely due to the production of various peptides and proteins with antimicrobial effects. In addition, *Bacillus* species play a role in enhancing plant growth by producing phytohormones, fixing nitrogen, and solubilizing phosphate [5–7]. Among these, *Bacillus velezensis*, originally identified as *Bacillus amyloliquefaciens* when it was first isolated from the Vélez River in Málaga, Spain, in 2005 [8], stands out for its ability to support plant growth. This is achieved through the production of compounds like indole acetic acid and siderophores, as well as a wide range of antimicrobial agents [7]. The effectiveness of *B. velezensis* in biocontrol is linked to its production of secondary metabolites, such as polyketides (including difcidin, bacillaene, and macrolactin) and cyclic lipopeptides (such as surfactin, fengycin, bacillibactin, iturin, and bacillomycin). These compounds not only foster plant growth but also offer protection against pathogens and trigger systemic resistance in plants [6,9].

Significant research has focused on investigating the bioactive potential of various *Bacillus velezensis* isolates [7,10–17]. Among these, *Bacillus velezensis* FZB42 stands out as one of the most thoroughly studied strains, recognized for its ability to produce an array of beneficial bioactive compounds. Notably, approximately 10% of its genome is allocated to the synthesis of antibiotics, with thirteen gene clusters dedicated to producing structurally diverse lipopeptides, which play a critical role in suppressing fungal pathogens such as *Fusarium oxysporum* [7,17,18].

Recent research emphasizes the diverse beneficial traits of *Bacillus velezensis* strains isolated from various environments. For instance, *B. velezensis* TS5 has been shown to enhance gut health and increase antioxidant capacity in murine models [15]. Similarly, *B. velezensis* N23 functions as a biocontrol agent against plant pathogens, exhibiting antifungal activity and promoting plant health, positioning it as a viable alternative to chemical fungicides [17]. Additionally, B. *velezensis* L9 has demonstrated promise in bioremediation by effectively degrading the food-contaminating mycotoxin zearalenone (ZEN) [10]. Another notable advancement involves the cloning and purification of a novel xylanase from B. velezensis RB.IBE29, marking a significant step forward in wood degradation research [12].

Furthermore, *B. velezensis* A5, isolated from deep-sea sediments, has shown potential for biocontrol against Tobacco Bacterial Wilt (TBW) [13]. *B. velezensis* B31 exhibits resistance to fusaric acid and effectively combats *Fusarium oxysporum*, offering robust control against tomato fusarium wilt [14]. Moreover, *B. velezensis* P1 has been identified as a safe and eco-friendly alternative to synthetic pesticides, particularly in managing Aspergillus carbonarius in Chardonnay grapes, improving the aroma profile without compromising grape quality [11]. Lastly, *B. velezensis* KTA01, isolated from peach tree soil in Korea, has demonstrated promising biocontrol potential against *Botryosphaeria dothidea* KACC45481, the causative agent of peach tree gummosis [16].

As of June 2024, more than 800 complete genomes of *Bacillus velezensis* strains have been archived in the NCBI database. With the continual advancement of bacterial genome sequencing technology, the mechanisms through which certain *Bacillus* species inhibit pathogens are becoming more comprehensible. Whole genome sequencing serves as a powerful tool to elucidate the genetic underpinnings of these mechanisms, revealing the links between genotype and phenotype and offering critical insights into the genes involved in the synthesis of valuable secondary metabolites. In the context of our ongoing project investigating microbial diversity in desert rock varnish [2,19,20] to the best of our knowledge, no studies have comprehensively investigated the bioactive gene potential of microorganisms thriving in this unique environment. Desert varnish, an underexplored habitat, may harbor microbial communities with novel genes and secondary metabolites that could have significant applications in medicine and agriculture. Recognizing the need to explore this potential, this study focuses on *Bacillus velezensis* strain RVMD2, isolated from rock varnish in the Ma'an Desert, Jordan.

This study aims to determine the taxonomic identity and provide a comprehensive characterization of the phylogenetic, genomic, and taxonomic features of *B. velezensis* RVMD2. Specifically, we seek to establish its taxonomic placement through phylogenomic analysis and investigate genes related to antibiosis, secondary metabolite biosynthesis, and other distinctive traits. These features are compared with those from the broader *Bacillus* genus to assess its potential contributions to biotechnological applications. To achieve these goals, we employ a multiphase classification strategy that integrates whole-genome shotgun sequencing with rRNA gene amplicon analysis, providing a robust framework for understanding the strain's genomic potential.

## Method

### Sampling site characteristics and procedures

In November 2022, desert rock varnish samples were aseptically collected from a semi-arid region near Ma'an, Jordan (coordinates: 30.188836°N, 35.639121°E), characterized by an average annual rainfall of approximately 50 mm. Rocks with flat surfaces were selected, sealed in sterile aluminum foil, and transported to the laboratory. Under sterile conditions, a flame-sterilized coarse bit was used to grind the rock into varnish powder, which was stored at -20°C. No permits were required for the described study as the field site is publicly accessible, and no protected or endangered species were involved.

### Isolation of bacteria with antagonistic potential

A 0.1 g sample of powdered rock varnish was inoculated onto Luedemann medium (DSMZ 877). After 72 hours of incubation at 37°C, colonies displaying distinct shapes, colors, and significant inhibition of adjacent bacterial and fungal growth were isolated and streak-purified for further analysis.

### 16S rRNA gene sequencing and phylogeny

The isolated strain underwent DNA extraction using the G-spin Total DNA Extraction Mini Kit (iNtRON Biotechnology, Suwon, Korea). Resultant DNA then served as a template for 16S rDNA amplification via PCR. DNA extraction followed

the manufacturer's guidelines. The SSU rRNA gene was amplified using the bacterial forward primer 27F (3'-AGRGTTYGATYMTGGCTCAG-5') paired with the 1492R primer (5'-RGYTACCTTGTTACGACTT-3'). PCR product was purified using the PCR quick-spin PCR Product Purification Kit as per the manufacturer's instructions, and sequenced by MACROGEN (Korea) using the Sanger method.

The 16S rRNA gene sequences were subjected to analysis and cross-referenced with the EzTaxon database (www.EzBioCloud.net) [21]. A partial 16S rRNA gene sequence of 1,238 base pairs (bp) was successfully obtained from the RVMD2 strain. This sequence has been archived in the NCBI GenBank under the accession number PP942176. To construct a phylogenetic tree based on the 16S rRNA gene, we employed Protologger (www.protologger.de), and sequence alignment was performed using MUSCLE (v3.8.31) under default parameters. The phylogenetic tree was generated through FastTree (v2.1.7) using the GTR model. Taxonomic identification was conducted by comparing sequence identities with the closest relatives from the SILVA Living Tree Project, ensuring that only species with validly published names from the DSMZ nomenclature list were included to guarantee precise classification [22]. The tree was visualized using the Interactive Tree Of Life (ITOL) online tool (https://itol.embl.de) [23] and subsequently refined using Inkscape (v1.0).

## Whole Genome Sequencing: Assembly, Annotation, and Feature Analysis

As outlined in the protocol for Gram-positive bacteria, genomic DNA was extracted using the G-spin Total DNA Extraction Mini Kit (iNtRON Biotechnology, Suwon, Korea). The purity and integrity of obtained DNA were assessed using the Nabi-UV/Vis Nano Spectrophotometer from MicroDigital, South Korea.

The genome of strain RVMD2 was sequenced using Illumina NextSeq 2000 (PE 150 bp, 15M reads/sample) at EzBiome Inc., (Gaithersburg, Maryland, USA). Quality control of raw reads was checked with MultiQC v.1.11[24]. The reads were assembled *de novo* using SPAdes v.3.13.0. [25], CheckM v1.0.18 [26] and QUAST v4.4 [27].

The genome of the studied strain was comprehensively annotated using the NCBI Prokaryotic Genome Annotation Pipeline (PGAP) [28] and the Bacterial Bioinformatics Resource Center (BV-BRC) [29]. Additional genomic characteristics were explored with several complementary tools, including the Microbial Genomes Atlas (MiGA) webserver (www.microbial-genomes.org) [30], EzBiome Genome-ID (www.ezbiome.app) and Galaxy Protologger (www.protologger.de) [23]. A circular map of the genome was generated via the Proksee web tool (https://proksee.ca/) [31]. Specifically, carbohydrate-active enzymes were identified using the Carbohydrate-Active Enzymes Database (CAZy) through Protologger. Genes associated with heavy metal resistance and plant growth promotion were annotated through BV-BRC. Genomic islands were predicted with Island Viewer 4 (www.pathogenomics.sfu.ca/islandviewer/) [32] and further compared using IslandCompare (v1.0) (https://islandcompare.ca/) [33], while prophage regions were detected using PHASTEST (www.phastest.ca) [34]. The presence of potential secondary metabolite biosynthesis gene clusters was identified, annotated, and analyzed using AntiSMASH 7.01 web service (https://antismash.secondarymetabolites.org/) [35], with default settings except for Detection Strictness, which was adjusted to 'strict'.

## Genome-Wide Taxonomic Classification and Phylogenetic Analysis

The initial taxonomic classification of strain RVMD2 was determined through the whole genome-based bacterial identification service provided by EzBiome Genome-ID (www.ezbiome.app) [36], which utilizes average nucleotide identity (ANI) to compare similarity values with a reference database. To further validate the taxonomic placement of strain RVMD2, multiple genome-scale methods were employed. Genome-wide GBDP tree analysis and digital DNA-DNA hybridization (dDDH) were conducted using the TYGS platform (https://tygs.dsmz.de/) [37]. This was complemented by an Average Amino Acid Identity (AAI) analysis, performed via the Microbial Genomes Atlas (MiGA) webserver (www.microbial-genomes.org) [30]. Additionally, the taxonomic assignment was verified using the GTDB-Tk r89 and ANI (FastANI) through Galaxy Protologger (www.protologger.de) [22]. The ANI values were visualized as a heatmap, generated using TBtools-I [38].

## Comparative genome analysis and pan-genome analysis

The Integrated Pan-Genome Analyser (IPGA) (https://nmdc.cn/ipga/), a web-based service for analyzing, comparing, and visualizing pan-genomes and individual genomes, has been utilized for genome analyses [39].Thanks to IPGA's substantial capabilities, 869 genomes of different *Bacillus velezensis* strains available in the NCBI database as of June 26, 2024, were reviewed. Out of these, 614 genomes were accepted and phylogenetically analyzed by IPGA. Based on the analysis results, the closest strains to RVMD2 were identified. The 14 closest strains to *Bacillus velezensis* strain RVMD2 (GCA_000973585.1, GCA_001709115.1, GCA_001723375.1, GCA_002082365.1, GCA_004337655.1, GCA_006350975.1, GCA_013122275.1, GCA_014204475.1, GCA_017599365.1, GCA_018398955.1, GCA_018771665.1, GCA_023614465.1, GCA_904841115.1, and GCA_904842145.1) were selected based on initial phylogenetic analysis using the (IPGA). These strains, along with RVMD2, underwent further pan-genome profiling to identify core and unique genes. COG annotation was performed to categorize these genes, and a phylogenetic tree was constructed to illustrate the number of shared gene clusters and visualize genetic relationships.

The closest three strains (Q12, CFSAN034340, and ASM-2) were selected for further genome comparison. Detailed visual fast whole-genome similarity analysis was performed using the FastANI 1.3.3 tool [40] available on the Proksee website [31]. Orthologous clusters were identified and annotated using the OrthoVenn3 online service (https://orthovenn3.bioinfotoolkits.net/) [41], which displayed the distribution of shared gene families among *Bacillus velezensis* strain RVMD2 and these three closest strains.

## Results and discussion

### 16S rRNA gene sequencing and phylogeny

In the framework of our research on microbial biodiversity associated with extreme desert habitats, a bacterium designated RVMD2, exhibiting distinct antagonistic activity against adjacent bacterial and fungal growth, was isolated from a desert rock varnish sample. An initial identification was conducted using 16S rRNA gene sequencing. The obtained partial 16S rRNA gene sequence (1,238 bp) of strain RVMD2, submitted to GenBank with the accession number PP942176. Top hits from 16S rRNA gene sequencing analysis of bacterial isolate RVMD2 are summarized in (Table 1), using the

**Table 1. A summary of top hits from 16S rRNA gene sequencing analysis of bacterial isolate RVMD2 in EzBioCloud [21] PP942176.**

| Rank | Name | Strain | Accession | Pairwise Similarity(%) | Completeness(%) |
|------|------|--------|-----------|------------------------|-----------------|
| 1 | *Bacillus velezensis* | CR-502 | AY603658 | 99.84 | 95.4 |
| 2 | *Bacillus siamensis* | KCTC 13613 | AJVF01000043 | 99.84 | 100 |
| 3 | *Bacillus amyloliquefaciens* | DSM 7 | FN597644 | 99.76 | 100 |
| 4 | *Bacillus subtilis* | NCIB 3610 | ABQL01000001 | 99.68 | 100 |
| 5 | *Bacillus nakamurai* | NRRL B-41091 | LSAZ01000028 | 99.68 | 100 |
| 6 | *Bacillus nematocida* | B-16 | AY820954 | 99.68 | 100 |
| 7 | *Bacillus cabrialesii* | TE3 | MK462260 | 99.51 | 100 |
| 8 | *Bacillus inaquosorum* | KCTC 13429 | AMXN01000021 | 99.51 | 100 |
| 9 | *Bacillus stercoris* | JCM 30051 | MN536904 | 99.51 | 100 |
| 10 | *Bacillus vallismortis* | DV1-F-3 | JH600273 | 99.43 | 100 |
| 11 | *Bacillus tequilensis* | KCTC 13622 | AYTO01000043 | 99.43 | 100 |
| 12 | *Bacillus rugosus* | SPB7 | JABUXO010000041 | 99.43 | 100 |
| 13 | *Bacillus atrophaeus* | JCM 9070 | AB021181 | 99.35 | 100 |
| 14 | *Bacillus halotolerans* | ATCC 25096 | LPVF01000003 | 99.35 | 100 |
| 15 | *Bacillus spizizenii* | NRRL B-23049 | CP002905 | 99.35 | 100 |

EzBioCloud [21]. This table ranks fifteen species based on their pairwise similarity to RVMD2, highlighting both the species name and their respective strain name and accession number. The results indicate a high degree of similarity between the bacterial isolate RVMD2 and several Bacillus species. The highest pairwise similarity, recorded at 99.84%, was found with both *Bacillus velezensis* (strain CR-502) and *Bacillus siamensis* (strain KCTC 13613), followed by *Bacillus amyloliquefaciens* (strain DSM 7) with a similarity of 99.76%. These findings, alongside other related *Bacillus* species detected in the analysis, indicate that isolate RVMD2 is positioned within a specific phylogenetic cluster of *Bacillus* species. This cluster comprises closely related species that are difficult to differentiate based solely on phenotypic traits or 16S rRNA gene phylogenetic analysis, owing to the gene's conserved nature [42]. It is important to note that *Bacillus amyloliquefaciens*, *Bacillus velezensis*, and *Bacillus siamensis* form a subgroup referred to as the "Operational Group *B. amyloliquefaciens*." This subgroup includes species commonly associated with soil and plants, reflecting shared genomic adaptations [43].

Fig 1 presents the 16S rRNA gene-based phylogenetic tree for strain RVMD2, generated using Protologger and cross-referenced with the SILVA Living Tree Project. The tree includes species with validly published names as recognized by the DSMZ nomenclature list. This phylogenetic analysis visually demonstrates the genetic relationships of strain RVMD2 with other *Bacillus* species. Notably, RVMD2 clusters closely with *Bacillus velezensis* and other members of the "Operational Group *B. amyloliquefaciens*" (including *B. siamensis* and *B. amyloliquefaciens*), highlighting their strong genetic similarity. These results align with the 16S rRNA gene sequencing analysis, where *B. velezensis*, *B. siamensis*, and *B. amyloliquefaciens* were identified as the most closely related species to RVMD2.

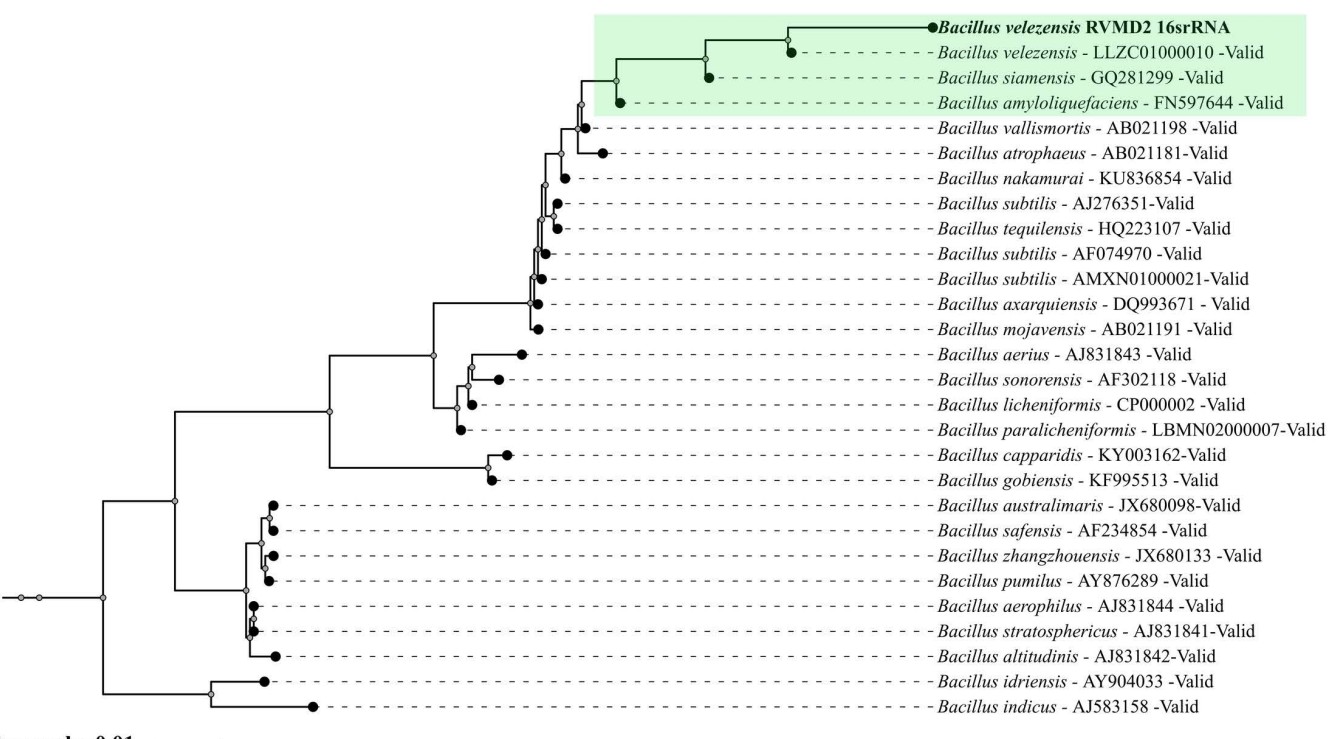

**Fig 1. Phylogenetic tree based on the 16S rRNA gene sequence of strain RVMD2, generated with Protologger, showcasing its closest relatives from the SILVA Living Tree Project. The tree includes species with validly published names as listed by DSMZ nomenclature. The visualization was further refined using the Interactive Tree of Life (iTOL) v5 tool [23].** The scale bar represents a genetic distance of 0.01 substitutions per site.

## Genomic features and annotations

The comprehensive analysis of *Bacillus velezensis* strain RVMD2's genomic characteristics, as outlined in Table 2 and Fig 2, was achieved by integrating data from several bioinformatics platforms, including MiGA, BV-BRC, NCBI PGAP, and Galaxy

Protologger. The genome comprises 112 contigs, spanning a total of 4,212,579 base pairs, with an N50 value of 897,830. The largest contig within the genome measures 1,046,988 base pairs. This integration provided a thorough and precise characterization of the strain's genomic structure.

Factors such as the initial material, sequencing technique, and sequence quality influenced the number and lengths of the contigs produced [46]. It is important to note that the circular genome map in Fig 2 is a visualization of this fragmented genome assembly (112 contigs) and does not represent a fully circularized chromosome. Such representations

**Table 2. Comprehensive Genomic Features of strain RVMD2. Data were compiled from multiple sources: The Microbial Genomes Atlas (MiGA) webserver [31], Bacterial Bioinformatics Resource Center (BV-BRC) (PATRIC) [44], NCBI Prokaryotic Genome Annotation Pipeline (PGAP) [28], functional analysis via Galaxy Protologger [45].**

| Category | Feature | Value | Source(s) |
|---|---|---|---|
| Genome Assembly | Contigs | 112 | BV-BRC |
| | Total length | 4,212,579 *bp* | BV-BRC |
| | N50 | 897,830 | MiGA, BV-BRC |
| | Longest sequence | 1046988 | MiGA |
| | G+C content | 45.94 | MiGA, BV-BRC |
| | G-C skew | -0.2443% | MiGA |
| | A-T skew | -0.2784% | MiGA |
| Genome Quality | Completeness | 99.59 | Protologger |
| | Contamination | 0.98 | Protologger |
| | Coarse Consistency | 99.0 | BV-BRC |
| | Fine Consistency | 97.1 | BV-BRC |
| Gene Prediction | Predicted proteins | 4,250 | MiGA |
| | Average protein length | 294.4216 aa | MiGA |
| | Coding density | 88.8913% | MiGA |
| | CDS | 4,230 | NCBI PGAP |
| | tRNA | 77 | BV-BRC |
| | rRNA | 4 | BV-BRC |
| | Genes (total) | 4,317 | NCBI PGAP |
| | Genes (coding) | 4,156 | NCBI PGAP |
| | Genes (RNA) | 87 | NCBI PGAP |
| | Pseudo Genes (total) | 74 | NCBI PGAP |
| Essential Genes | Essential genes found | 105/106 | MiGA |
| | Multiple Copies | (**2**) Ribosomal_L6, (**2**) tRNA-synt_1d, (**2**) Methyltransf_5, (**2**) GTPase Era, (**2**) *gyrA*. | MiGA |
| | Missing Genes | *rpmH* | MiGA |
| | UBCG* Recovery | 92/92 | EzBiome Genome-ID |
| Functional Analysis | Number of transporters | 209 | Protologger |
| | Number of secretion genes | 26 | Protologger |
| | Number of unique enzymes | 898 | Protologger |

*UBCG: Universal Bacterial Core Gene

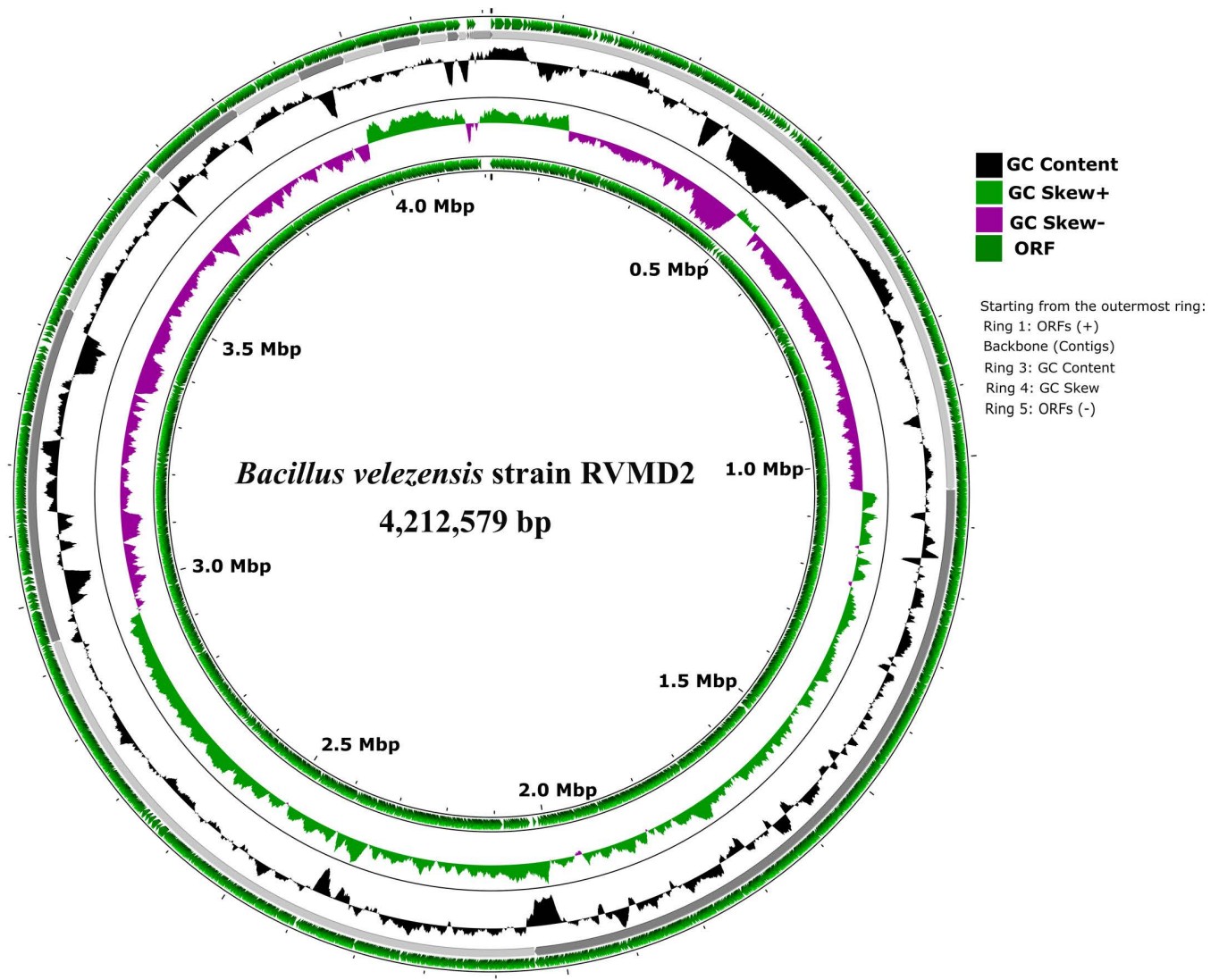

**Fig 2. Circular genome map of *Bacillus velezensis* strain RVMD2 generated through Proksee web-based tool.** The tracks display open reading frames (ORFs) on forward and reverse strands, backbone contigs, GC content, and GC skew (green/purple graph), illustrating the detailed genomic structure. The outermost ring shows ORFs on the positive strand, followed by backbone contigs. The third ring represents GC content with black peaks for higher GC areas, and the fourth ring shows GC skew, with green for positive (GC Skew+) and purple for negative (GC Skew-). The innermost ring shows ORFs on the negative strand.

are widely used in bacterial genome studies to facilitate the interpretation of genomic features. The G+C content is 45.94%. However, the genome size and G+C content values fall within the range observed for other related *Bacillus* species, as detailed in S2 Table. Quality metrics indicate a high-quality genome assembly, with completeness at 99.59%, contamination at 0.98%, and consistency scores (coarse and fine) at 99.0 and 97.1, respectively. These metrics meet the minimum standards for taxonomic applications [47]. The genome also includes 77 tRNA genes and 4 rRNA genes, which are critical for protein synthesis. Gene prediction analysis reveals a substantial proteome, with 4,250 predicted proteins averaging 294.42 amino acids in length. The coding density is 88.89%, indicating an efficient genomic coding capability. Isolated from a harsh desert environment, strain RVMD2 has 105 out of 106 essential genes, with some genes duplicated,

highlighting its evolutionary adaptations for resilience and survival [48]. Functional analysis identified a significant number of transporters (209), secretion genes (26), and unique enzymes (898). Additionally, the UBCG recovery rate for this genome was 92 out of 92, demonstrating the presence of all Universal Bacterial Core Genes, which further supports the quality and completeness of the genome assembly.

The genome of strain RVMD2 has been also functionally annotated to reveal a detailed distribution of its genes across various biological subsystems. According to the Sankey diagram (Fig 3A), a total of 1,822 genes are categorized into 278 subsystems. A significant portion of these genes is dedicated to metabolic functions, with 747 genes spread over 92 subsystems. Other notable categories include protein processing (224 genes in 41 subsystems), stress response, defense, and virulence (137 genes in 34 subsystems), and cellular processes (242 genes in 29 subsystems). Additionally, Fig 3B provides a summary of protein annotations. It highlights that out of the total proteins, 1,010 are hypothetical, while 3,455 have functional assignments. Within the functionally annotated proteins, 1,047 have Enzyme Commission (EC) numbers, 876 are assigned Gene Ontology (GO) terms, and 774 are mapped to KEGG pathways. The PATRIC annotation further reveals 4,110 proteins classified under genus-specific protein families (PLFams) and 4,213 proteins associated with cross-genus protein families (PGFams). This comprehensive annotation underscores the significant challenge posed by the large number of functionally uncharacterized genes in *Bacillus velezensis* RVMD2. Given the huge ability of this bacterium to produce bioactive substances, these uncharacterized genes offer substantial opportunities for future exploration to uncover the functions of these hypothetical proteins and their roles in microbial [49].

Table 3 outlines the antibiotic resistance genes found in the genome. The aminoglycoside resistance gene (*ant(6)*) is located on contig 6, reverse direction, showing 63.73% identity and 99.3% reference coverage. Two beta-lactam resistance genes (*bla*) were identified on contig 3: WP_011053164.1 (64.69% identity, 98.06% coverage) and WP_063839879.1 (46.12% identity, 100% coverage), both in the forward direction. The *clbA* gene, resistant to lincosamide, macrolide, and streptogramin, is on contig 4, reverse direction, with 97.42% identity and 100% coverage. The macrolide resistance gene (*abc-f*) is on contig 4, forward direction, showing 70.98% identity and 98.72% coverage. The streptothricin resistance gene (*satA*) and tetracycline resistance gene (*tet*) are both on contig 1, reverse direction, with 79.77% and 86.43% identity, and 100% and 99.78% coverage, respectively. The presence of complete AMR genes does not necessarily indicate a resistant phenotype [50].

## Whole genome-based taxonomic placement and phylogenetic analyses

Expanding on the initial findings that indicated the RVMD2 genome was part of the Operational Group *Bacillus amyloliquefaciens* based on the 16S rRNA gene sequence, a more thorough examination was conducted using whole-genome analysis. This step was prompted by growing evidence that the 16S rRNA gene sequence has limited resolution for distinguishing closely related species within certain genera [51–53]. The subsequent analysis utilized EzBiome Genome-ID [36], to perform an in-depth comparison using Average Nucleotide Identity (ANI) against a reference database. Additionally, The Type (Strain) Genome Server (TYGS) was employed to enhance species descriptions, genome-based phylogenies, and (sub-)species delineation through digital DDH. The Protologger tool was also used for comprehensive genome description. This multifaceted approach aimed to definitively clarify and confirm the precise taxonomic classification of strain RVMD2.

Based on Table 4, the analysis using the EzBiome Genome-ID tool identified *Bacillus velezensis* as the top match for bacterial isolate RVMD2 with an ANI of 97.803%, 16S similarity of 99.92%, *recA* identity of 98.563%, *rplC* identity of 99.524%, Mash identity of 97.8%, and ANI coverage of 91.1648%. The ANI percentage (97.803%) is higher than the species demarcation threshold of 95–96%[47]. *B. siamensis* followed closely with an ANI of 94.432%, 16S similarity of 99.92%, *recA* identity of 96.456%, *rplC* identity of 99.048%, Mash identity of 95.24%, and ANI coverage of 88.4169%. Further, *B. amyloliquefaciens* showed an ANI of 94.1522%, 16S similarity of 99.84%, *recA* identity of 97.701%, *rplC* identity of 99.206%, Mash identity of 94.56%, and ANI coverage of 84.1301%, supporting the findings from the initial 16S rRNA gene

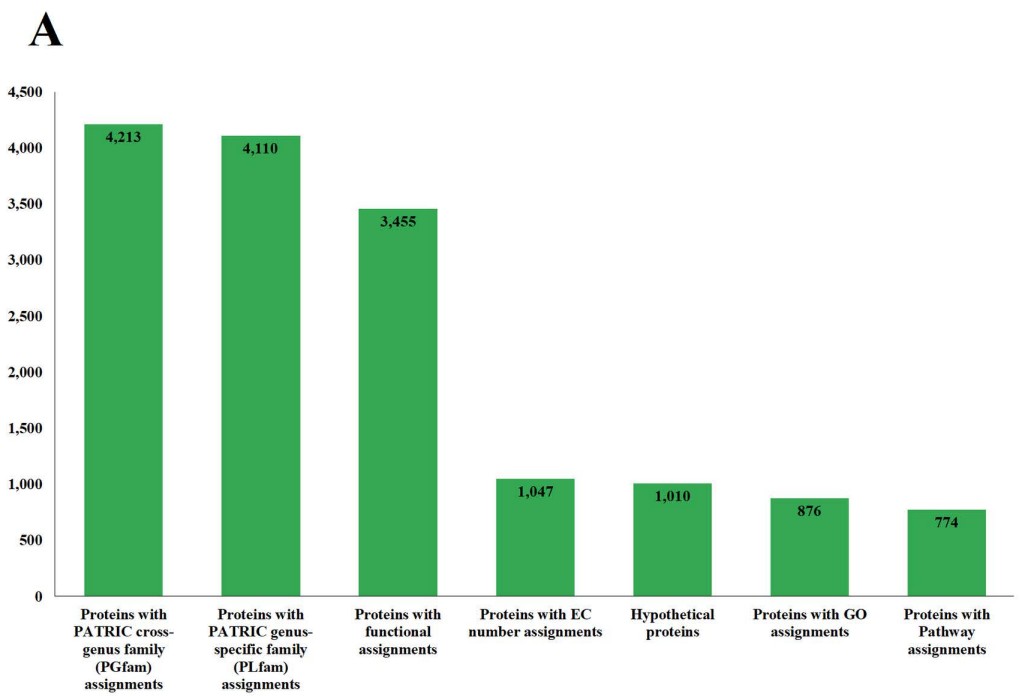

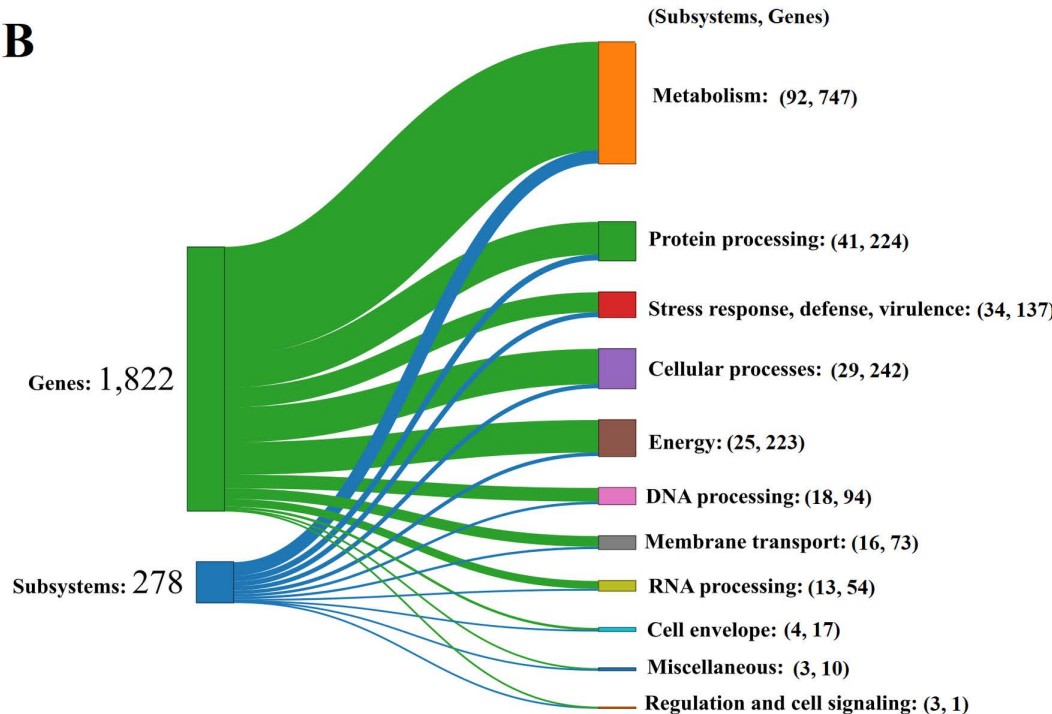

**Fig 3. Functional Characterization of the *Bacillus velezensis* RVMD2 Genome.** (A) Sankey diagram illustrating the distribution of genes into sub-systems classified by biological functions. (B) Summary of protein annotations detailing cross-genus and genus-specific family assignments, pathway connections, GO categorizations, EC number designations, and other functional annotations, along with hypothetical proteins—all as annotated by BV-BRC [44].

**Table 3. Antibiotic resistance determinants in bacterial isolate RVMD2 as identified by whole genome analysis using EzBiome Genome-ID.**

| Class | Subclass | Gene | Accession | Con-tig | Location | Dir. | Iden. (%) | Ref. cov. (%) |
|---|---|---|---|---|---|---|---|---|
| Aminoglycoside | Streptomycin | *ant(6)* | WP_087343787.1 | 6 | 45,824..46,693 | **Reverse** | 63.73 | 99.3 |
| Beta-lactam | Beta-lactam | *bla* | WP_011053164.1 | 3 | 343,173..344,087 | **Forward** | 64.69 | 98.06 |
| Beta-lactam | Beta-lactam | *bla* | WP_063839879.1 | 3 | 284,047..284,796 | **Forward** | 46.12 | 100 |
| Lincosamide, Macro-lide, Streptogramin | Lincosamide, Macro-lide, Streptogramin | *clbA* | WP_012116915.1 | 4 | 510,102..511,151 | **Reverse** | 97.42 | 100 |
| Macrolide | Macrolide | *abc-f* | WP_003234144.1 | 4 | 308,898..32,526 | **Forward** | 70.98 | 98.72 |
| Streptothricin | Streptothricin | *satA* | WP_003242546.1 | 1 | 941,125..941,649 | **Reverse** | 79.77 | 100 |
| Tetracycline | Tetracycline | *tet* | WP_003242953.1 | 1 | 939,540..940,916 | **Reverse** | 86.43 | 99.78 |

**Table 4. Top average nucleotide identity (ANI) hits for bacterial isolate RVMD2 whole genome: A detailed comparative analysis using EzBiome Genome-ID tool, focusing on ANI, 16S Similarity (%), *recA* Identity (%), *rplC* Identity (%), and Mash Identity (%) among closely related *Bacillus* species.**

| Rank | Hit Taxon | ANI(%) | 16S Similarity(%) | *recA* Identity(%) | *rplC* Identity(%) | Mash Identity(%) | ANI Coverage(%) |
|---|---|---|---|---|---|---|---|
| 1 | ***Bacillus velezensis*** | 97.803 | 99.92 | 98.563 | 99.524 | 97.8 | 91.1648 |
| 2 | *B. siamensis* | 94.432 | 99.92 | 96.456 | 99.048 | 95.24 | 88.4169 |
| 3 | *B. amyloliquefaciens* | 94.1522 | 99.84 | 97.701 | 99.206 | 94.56 | 84.1301 |
| 4 | *Bacillus subtilis* subsp. *subtilis* | 84.224 | 99.76 | N/A | 94.762 | N/A | 23.4812 |
| 5 | *Bacillus nakamurai* | 87.0542 | 99.76 | 92.433 | 97.778 | 89.31 | 76.205 |
| 6 | *Bacillus subtilis* subsp. *stercoris* | 84.06 | 99.67 | N/A | N/A | 84.88 | 23.8463 |
| 7 | *Bacillus subtilis* subsp. *inaquosorum* | 84.3836 | 99.61 | 84.909 | 94.444 | N/A | 24.1974 |
| 8 | *Bacillus cabrialesii* | 84.2093 | 99.61 | 85.005 | 94.603 | N/A | 24.0418 |
| 9 | *Bacillus vallismortis* | 83.9848 | 99.53 | N/A | 95.079 | N/A | 22.404 |
| 10 | *Bacillus tequilensis* | 84.5846 | 99.53 | N/A | 94.603 | N/A | 24.6033 |
| 11 | *Bacillus subtilis* subsp. *spizizenii* | 84.0703 | 99.45 | N/A | 94.444 | N/A | 23.9936 |
| 12 | *Bacillus atrophaeus* | 83.7286 | 99.45 | N/A | 95.079 | N/A | 25.0541 |
| 13 | *Bacillus halotolerans* | 84.0344 | 99.45 | 85.673 | 95.079 | N/A | 24.7541 |
| 14 | *Bacillus mojavensis* | 83.8931 | 99.37 | 86.151 | 94.762 | N/A | 23.976 |
| 15 | *Bacillus licheniformis* | 84.2518 | 98.5 | N/A | N/A | N/A | 5.5703 |
| 16 | *Bacillus haynesii* | 84.441 | 98.26 | N/A | 91.111 | N/A | 5.1897 |

sequencing analysis. Other species, including *Bacillus nakamurai*, *B. subtilis* subsp. *stercoris*, and *B. cabrialesii*, showed ANI values ranging from 83.7286% to 87.0542%.

Analysis via MiGA using AAI, as detailed in S1 Table, identified *B. velezensis* NZ CP036527 (99.86% ANI) and NZ CP010556 (99.35% ANI) as the closest relatives. In S2 Table, pairwise dDDH values calculated using TYGS showed that strain RVMD2 has high dDDH values with *B. velezensis* NRRL B-41580 (80.2%), *B. amyloliquefaciens* subsp. *plantarum* FZB42 (80%), and *B. methylotrophicus* KACC 13105 (79.5%), all above the 70% threshold [47]. Minimal G+C content differences confirm their close genetic relationship. Consequently, we designated it as *B. velezensis* RVMD2, aligning with the reclassification of *B. methylotrophicus* KACC 13105 and *B.amyloliquefaciens* subsp. *plantarum* FZB42 as synonyms of *B. velezensis* NRRL B-41580 [54]. Similarly, the whole genome-based phylogeny, as shown in Fig 4, demonstrated that *B. velezensis* strain RVMD2 formed a clade with several *B. velezensis* species, including the type strain *B. velezensis* NRRL B-41580T (also shown in the ANI values heatmap in Fig 5). The phylogenetic tree places *B. velezensis* strain

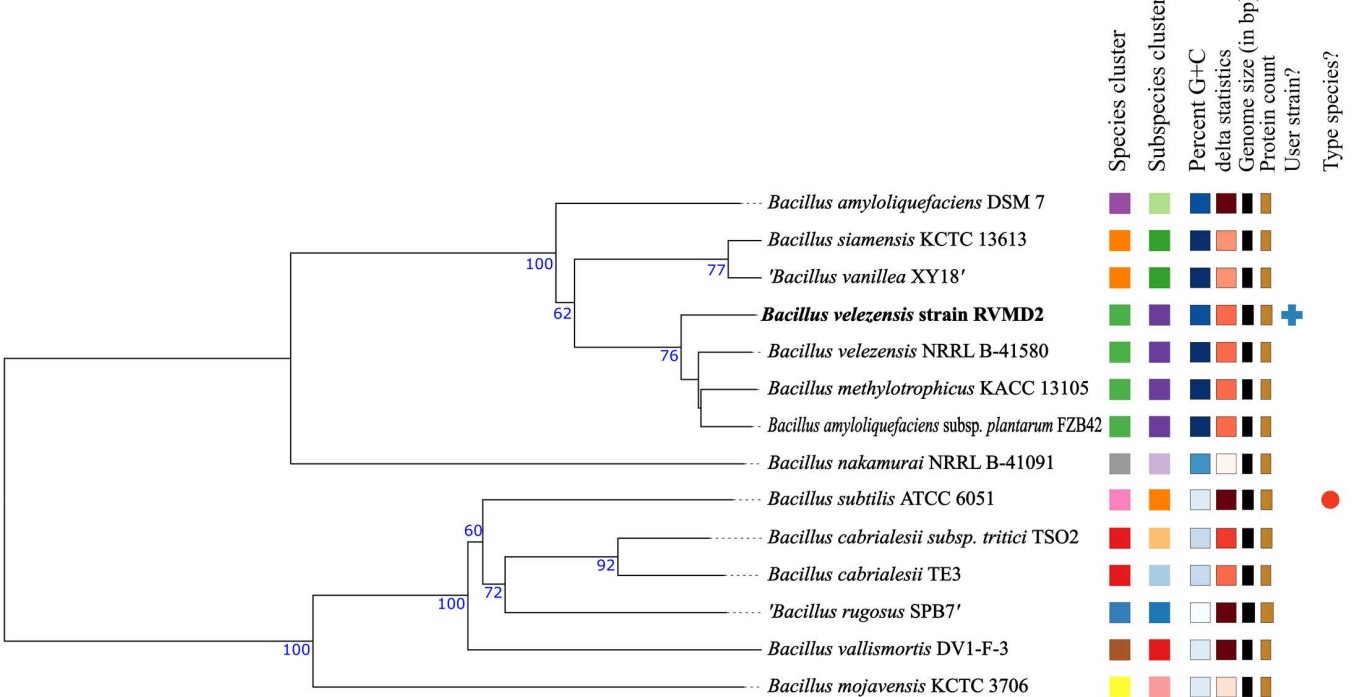

**Fig 4. Phylogenetic analysis of *Bacillus* species, including *B. velezensis* RVMD2, based on whole-genome comparisons using the Genome Blast Distance Phylogeny (GBDP) method, conducted via the TYGS server.** The tree was constructed with FastME 2.1.6.1 [55] using GBDP distances derived from genome sequences, with branch lengths reflecting the GBDP distance metric (d5). Numbers above branches represent GBDP pseudo-bootstrap support values exceeding 60% from 100 replications, with an average branch support of 75.6%. The tree was midpoint-rooted [56]. Leaf labels indicate species and subspecies clusters based on digital DNA-DNA hybridization (dDDH), genomic G+C content, delta values, genome length, and protein count. Lower delta values correspond to higher accuracy in tree-likeness [57].

RVMD2 in a well-supported clade with high bootstrap values, confirming its close relationship with other *B. velezensis* strains.

Analysis of CAZyme, Heavy Metal, Plant Growth-Promoting Genes, Genomic Islands, and Prophage Sequences in *Bacillus velezensis* RVMD2

The genome of *B. velezensis* RVMD2 has been annotated with 324 genes, as shown in Fig 6, encoding carbohydrate-active enzymes (CAZymes), including 117 glycosyl transferases (GTs), 103 glycoside hydrolases (GHs), 75 carbohydrate-binding modules (CBMs), 26 carbohydrate esterases (CEs), and three polysaccharide lyases (PLs). These CAZymes, cataloged in the CAZy database, play crucial roles in the synthesis of secondary metabolites and exhibit anti-microbial properties by targeting fungal cell walls and lysing bacteria to control pathogen growth [58,59].

As listed in S3 Table, the GH family enzymes, including GH1, GH3, GH16, GH30, and GH51, enable *B. velezensis* RVMD2 to efficiently degrade cellulose and hemicellulose [60]. The diverse array of cellulase and hemicellulase genes suggests a significant role in promoting plant growth by decomposing organic matter and releasing nutrients into the soil. The presence of GH18 and GH19 chitinases in *Bacillus velezensis* RVMD2 suggests significant antifungal properties, positioning it as an effective biocontrol agent against plant pathogens. Additionally, carbohydrate esterases, such as CE4 and CE7, involved in xylan degradation, further underscore its potential in promoting soil health [60]. The organism's repertoire of glycosyl transferases, carbohydrate-binding modules, and polysaccharide lyases enhances its capacity for synthesizing and degrading complex polysaccharides with high efficiency [61]. These findings indicate that *B. velezensis* RVMD2 holds considerable promise for agricultural applications, particularly in promoting plant growth and protecting

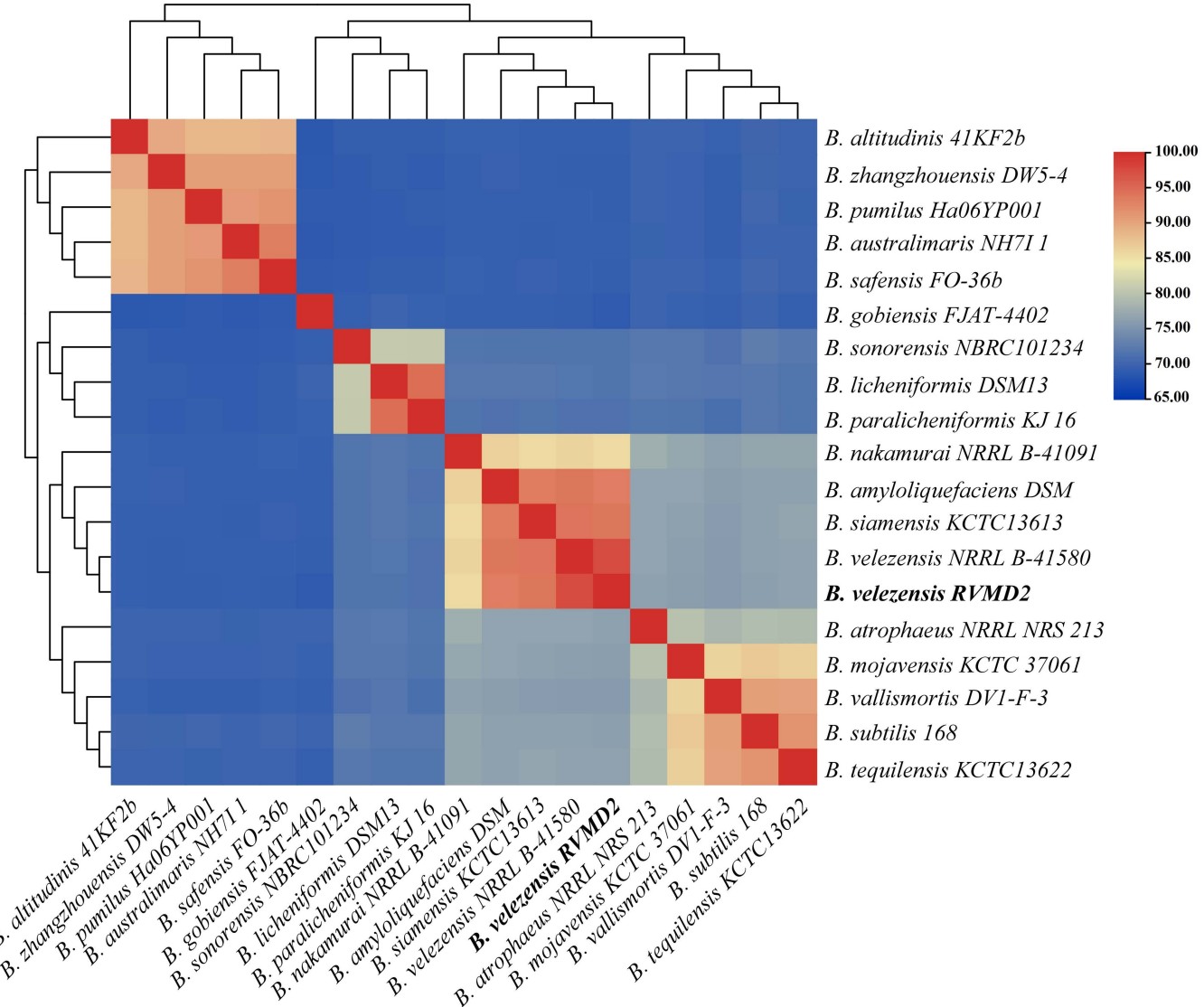

**Fig 5. ANI values (FastANI) heatmap of *B. velezensis* RVMD2 and its closest *Bacillus* strains according to Protologger, with hierarchical clustering.** The color gradient from red to blue indicates the degree of genomic similarity, with red representing higher similarity (closer species) and blue indicating lower similarity.

crops from fungal diseases, as well as in biotechnological and industrial contexts that demand efficient carbohydrate breakdown [58,59].

The genome annotation of *Bacillus velezensis* RVMD2, conducted through the Bacterial and Viral Bioinformatics Resource Center (BV-BRC), revealed numerous genes implicated in both heavy metal resistance and plant growth promotion. As presented in Table 5, the analysis identified 50 relevant genes. Among these, 11 genes are associated with heavy metal resistance, including P-type ATPase (1 gene), CzcD resistance protein (3 genes), chromate transport protein (2 genes), arsenical resistance operon repressor (2 genes), and lead/cadmium/zinc/mercury/copper-transporting ATPase (2 genes). The genome also contains 3 genes involved in 1-aminocyclopropane-1-carboxylate (ACC) biosynthesis, specifically 2 acyl-CoA dehydrogenase genes and 1 long-chain acyl-CoA dehydrogenase gene. Furthermore, 12 genes are

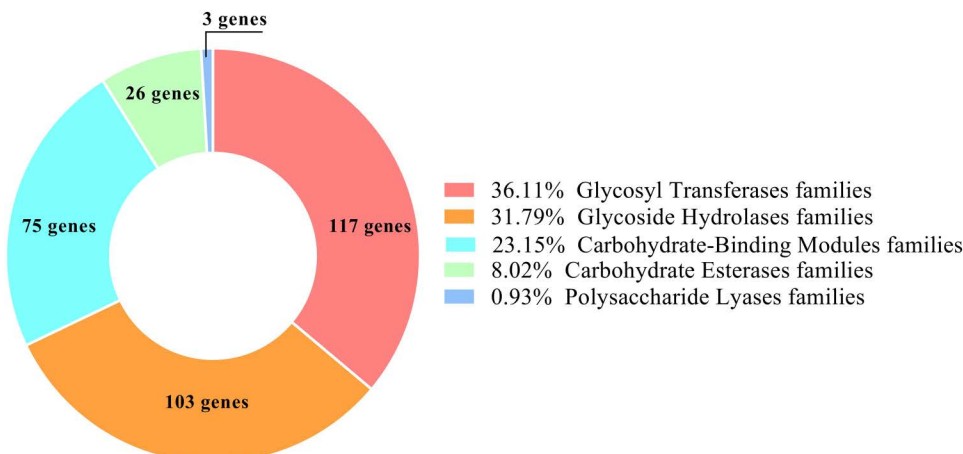

**Fig 6. Distribution of carbohydrate-active enzymes (CAZymes) identified in the genome of *Bacillus velezensis* RVMD2, as determined through analysis using Protologger v0.99 [ 22].**

responsible for indole-3-acetic acid (IAA) biosynthesis, including 3-dehydroquinate synthase (1 gene), chorismate mutase (1 gene), chorismate synthase (1 gene), and indole-3-glycerol phosphate synthase (1 gene). For inorganic phosphate solubilization, 4 genes were identified, including 4-hydroxy-3-methylbut-2-enyl diphosphate reductase (1 gene) and components of the Ktr potassium uptake system (3 genes). Additionally, 7 genes contribute to nitrate transport and reduction, involving nitrite reductase (2 genes) and nitrate/nitrite transporter NarT (1 gene). Iron uptake and siderophore production are supported by 12 genes, including ABC-type Fe3+ transport system (1 gene), siderophore transport protein (1 gene), several uncharacterized iron transporters (5 genes), 2,3-dihydroxybenzoate-AMP ligase (1 gene), and isochorismate synthase (1 gene). The presence of these genes equips *B. velezensis* RVMD2 with notable capabilities in heavy metal resistance and plant growth enhancement, rendering it a promising candidate for bioremediation and agricultural use [59,62].

Predicted by IslandViewer 4 tool, Fig 7A, the *B. velezensis* strain RVMD2 genome possesses a total of 13 genetic islands (GIs). These islands are located in different zones across the genome with lengths ranging from 2,544 bp to 142,704 bp and contain some genes of potential biotechnological, plant growth promotion, and secondary metabolite biosynthesis interest like 16S rRNA methyltransferase, alanine racemase, non-ribosomal peptide synthetase, collagen-like protein, and N-acetylmuramoyl-L-alanine amidase. For example, Non-ribosomal peptide synthetase (NRPS) synthesizes non-ribosomal peptides, including antibiotics and other bioactive compounds, crucial for discovering and producing novel natural products [13,63]. As visualized in Fig 7B, seven prophage elements in the *B. velezensis* strain RVMD2 genome, encompassing 308 phage genes, are predicted. The figure annotates phage-like proteins, attachment sites, and other relevant proteins, indicating their genomic positions. The genomic islands were identified using IslandViewer 4 [32], and the prophage regions were predicted using PHASTEST [34]. These predictions are based on sequence composition and annotations, and while the level of fragmentation in the genome assembly may have some influence, the results are still considered reliable and align with standard practices in bacterial genomics. S4 Table further describes these prophage regions, detailing their length (ranging from 31.6Kb to 87.4Kb), completeness, score, total proteins, position, most common phage, and GC content (ranging from 35.69% to 47.16%), illustrating the diversity and complexity of prophage regions within the genome. The genomic islands, which represent clusters of likely horizontally acquired genes, were analyzed by comparing the genome of strain RVMD2 with those of its closely related phylogenetic neighbors using IslandCompare [33]. This comparison revealed that strain RVMD2 differs from its phylogenetic neighbors in the number, type, and location of these genomic islands. Notably, RVMD2 contains two unique clusters that were not detected in

**Table 5. Heavy metal and plant growth-promoting genes in the *Bacillus velezensis* RVMD2 genome were obtained from the annotation using (BV-BRC).**

| Function | Start | End | Product |
|---|---|---|---|
| Heavy metal resistance | 700068 | 702179 | Cadmium, zinc and cobalt-transporting P-type ATPase (EC 3.6.3.3) (EC 3.6.3.5) |
| | 159157 | 160026 | Cobalt-zinc-cadmium resistance protein |
| | 115889 | 116782 | Cobalt-zinc-cadmium resistance protein |
| | 377680 | 378573 | Cobalt-zinc-cadmium resistance protein |
| | 55342 | 56283 | Cobalt/zinc/cadmium resistance protein *CzcD* |
| | 393178 | 393768 | Chromate transport protein |
| | 392645 | 393181 | Chromate transport protein |
| | 228703 | 229800 | Arsenical resistance operon repressor |
| | 172042 | 172392 | Arsenical resistance operon repressor |
| | 697489 | 699918 | Lead, cadmium, zinc and mercury transporting ATPase (EC 3.6.3.3) (EC 3.6.3.5); Copper-translocating P-type ATPase (EC 3.6.3.4) |
| | 504656 | 506569 | Lead, cadmium, zinc and mercury transporting ATPase (EC 3.6.3.3) (EC 3.6.3.5); CopFper-translocating P-type ATPase (EC 3.6.3.4) |
| ACC biosynthesis | 794673 | 795806 | Acyl-CoA dehydrogenase, short-chain specific (EC 1.3.8.1) |
| | 778739 | 780523 | Long chain acyl-CoA dehydrogenase [*fadN-fadA-fadE* operon] (EC 1.3.8.8) |
| | 117335 | 118480 | Acyl-CoA dehydrogenase |
| Indole-3-acetic acid biosynthesis | 598042 | 599130 | 3-dehydroquinate synthase (EC 4.2.3.4) |
| | 597662 | 598045 | Chorismate mutase II (EC 5.4.99.5) |
| | 599130 | 600302 | Chorismate synthase (EC 4.2.3.5) |
| | 594158 | 594910 | Indole-3-glycerol phosphate synthase (EC 4.1.1.48) |
| | 49393 | 50277 | Aminodeoxychorismate lyase (EC 4.1.3.38) |
| | 595891 | 597438 | Anthranilate synthase, aminase component (EC 4.1.3.27) |
| | 594903 | 595919 | Anthranilate phosphoribosyltransferase (EC 2.4.2.18) |
| | 591527 | 592324 | Tryptophan synthase alpha chain (EC 4.2.1.20) |
| | 592317 | 593519 | Tryptophan synthase beta chain (EC 4.2.1.20) |
| | 245972 | 246964 | Tryptophanyl-tRNA synthetase (EC 6.1.1.2) |
| | 129088 | 129606 | Substrate-specific component TrpP of tryptophan ECF transporter |
| | 378653 | 378817 | Tryptophan RNA-binding attenuator protein-inhibitory protein anti-TRAP |
| Inorganic phosphate solubilization | 878293 | 879237 | 4-hydroxy-3-methylbut-2-enyl diphosphate reductase (EC 1.17.7.4) |
| | 947710 | 948378 | *KtrAB* potassium uptake system, peripheral membrane component KtrA |
| | 473713 | 475065 | *KtrCD* potassium uptake system, integral membrane component KtrD |
| | 565797 | 566462 | *KtrCD* potassium uptake system, peripheral membrane component KtrC |
| Nitrate transport and reduction | 311411 | 311731 | Nitrite reductase [NAD(P)H] small subunit (EC 1.7.1.4) |
| | 308973 | 311390 | Nitrite reductase [NAD(P)H] large subunit (EC 1.7.1.4) |
| | 288129 | 289319 | Nitrate/nitrite transporter *NarT* |
| | 296845 | 297402 | Respiratory nitrate reductase delta chain (EC 1.7.99.4) |
| | 291680 | 295366 | Respiratory nitrate reductase alpha chain (EC 1.7.99.4) |
| | 295356 | 296819 | Respiratory nitrate reductase beta chain (EC 1.7.99.4) |
| | 297399 | 298070 | Respiratory nitrate reductase gamma chain (EC 1.7.99.4) |

*(Continued)*

**Table 5.** (Continued)

| Function | Start | End | Product |
|---|---|---|---|
| Iron uptake and siderophore production | 764490 | 765314 | ABC-type Fe3+-siderophore transport system, ATPase component |
| | 849701 | 851029 | Siderophore transport protein |
| | 242258 | 243211 | Uncharacterized iron compound ABC uptake transporter, permease protein |
| | 239603 | 240547 | Uncharacterized iron compound ABC uptake transporter, substrate-binding protein |
| | 240569 | 241327 | Uncharacterized iron compound ABC uptake transporter, ATP-binding protein |
| | 241321 | 242268 | Uncharacterized iron compound ABC uptake transporter, permease protein |
| | 853972 | 855597 | 2,3-dihydroxybenzoate-AMP ligase [bacillibactin] siderophore biosynthesis |
| | 852757 | 853953 | Isochorismate synthase [bacillibactin] siderophore biosynthesis |
| | 856556 | 863683 | Siderophore biosynthesis non-ribosomal peptide synthetase modules, Bacillibactin synthetase |
| | 855615 | 856541 | Isochorismatase [bacillibactin] siderophore biosynthesis |
| | 851948 | 852733 | 2,3-dihydro-2,3-dihydroxybenzoate dehydrogenase [siderophore biosynthesis] |
| | 850943 | 851812 | Trilactone hydrolase [bacillibactin] siderophore |
| | 112404 | 113693 | Siderophore biosynthesis protein, monooxygenase |

other closely related strains (S1 Fig). The presence of these diverse and functionally significant genes within the genomic islands and prophage sequences suggests genomic regions acquired through horizontal gene transfer, which may contribute to the unique capabilities and adaptations of strain RVMD2 [63].

**Identifying Secondary Metabolite Biosynthesis Gene Clusters for Biocontrol Attributes**

The analysis of the *B. velezensis* RVMD2 genome using antiSMASH, set at strict detection, revealed that approximately 17.85% (751,806 bp) of the genome is dedicated to the potential synthesis of secondary metabolites with antimicrobial activity. These metabolites include Bacillaene, Fengycin, Difficidin, Bacilysin, Bacillibactin, Macrolactin H, Surfactin, and Andalusicin, among several other compounds as summarized in Table 6. This strain exhibits a diverse array of biosynthetic gene clusters (BGCs) responsible for producing these secondary metabolites. Fig 8A illustrates the chemical structures of these metabolites, while Fig 8B provides a schematic representation of the gene clusters related to their biosynthesis. *Bacillus* species produce antimicrobial peptides via two pathways: non-ribosomal peptides (NRPs) by non-ribosomal peptide synthetases (NRPSs) and polyketide synthases (PKSs), resulting in antibiotics like surfactin and bacillibactin; and ribosomally synthesized and post-translationally modified peptides (RiPPs), such as bacteriocins, which disrupt pathogenic cell membranes or interfere with metabolism [6].

Among the twelve identified putative gene cluster regions, four encoded for NRPS (non-ribosomal peptide synthetase), three for transAT-PKS (trans-acyl transferase polyketide synthetase), two for terpene, one for other types, and two for T3PKS (Type III polyketide synthetase). The NRPS clusters showed 100% similarity to reported gene clusters for Bacillibactin and Fengycin, 82% for Surfactin, and 71% for Bacillaene. Fengycin is recognized for its antifungal effects, while Surfactin has antibacterial, antifungal, and antiviral potential [10,15]. Among the transAT-PKS clusters, the similarity with reported gene clusters for Difficidin, Macrolactin H, and Bacillaene were 100%, 100%, and 35%, respectively. Macrolactin, a 24-membered macrolide, effectively inhibits various Gram-positive pathogens [64]. The NRPS regions account

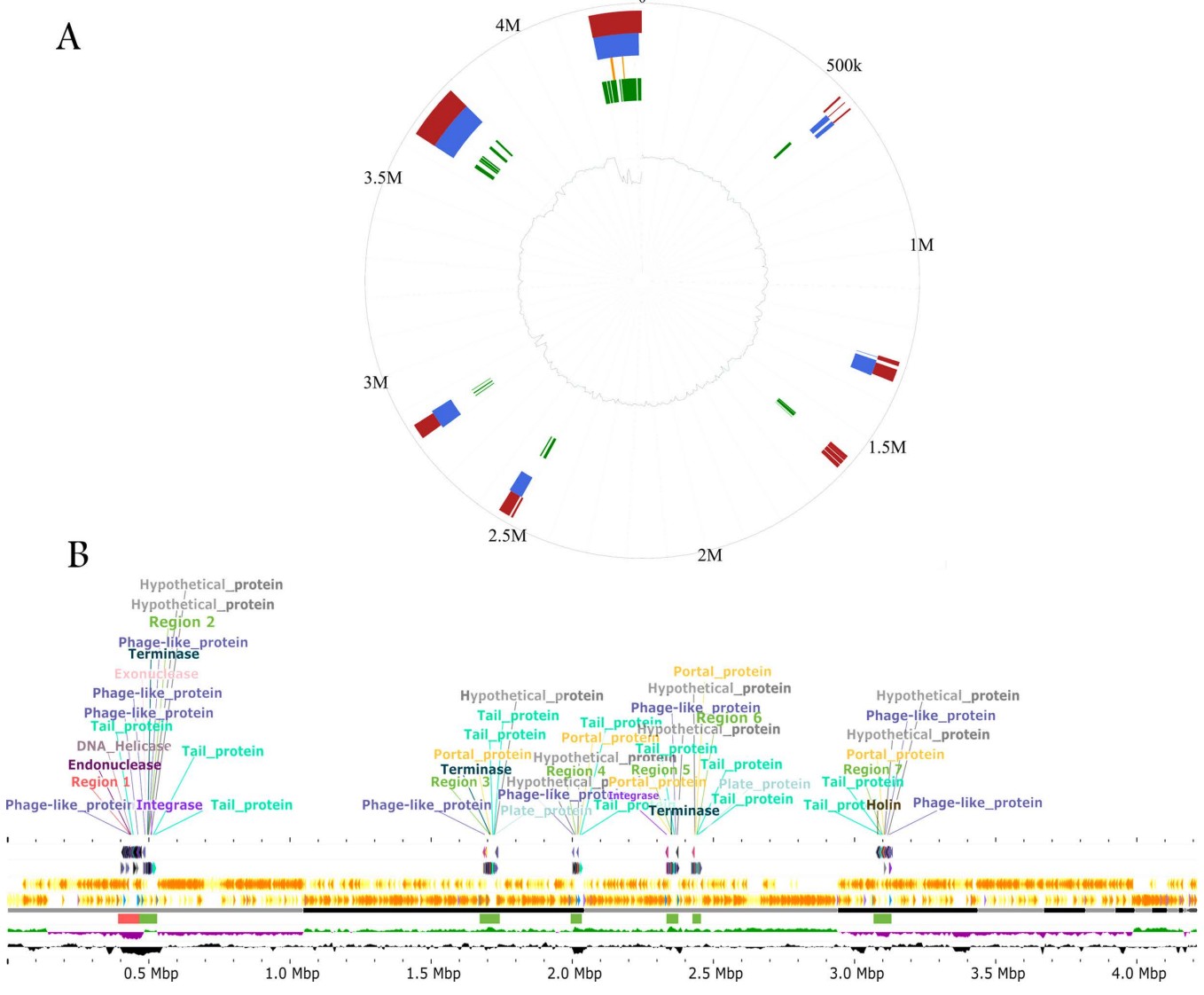

**Fig 7. Genomic islands and prophage regions in _B. velezensis_ RVMD2.** (A) Genomic islands in strain RVMD2 were predicted using IslandViewer 4. The circular plot illustrates these genomic islands, with the outer circle representing the genomic scale in Mbps. The genomic islands are depicted in different colors: red for integrated detection, blue for IslandPath-DIMOB, orange for SIGI-HMM, and green for IslandPick. (B)The linear genome viewer from PHASTEST [34] shows prophage regions in the strain RVMD2 genome, highlighting bacterial genes, attachment sites, phage-related proteins, and GC content. Key regions are annotated with hypothetical proteins, integrases, tail proteins, and other relevant phage-related proteins.

for approximately 9.38% of the total genome length in _B. velezensis_ RVMD2. This is significantly higher compared to _B. velezensis_ FZB42, where 8.5% of the genome is dedicated to non-ribosomal production of secondary metabolites, and more than double that of its closely related Gram-positive counterpart, _B. subtilis_ 168 [65].

Regions 1.3, 1.4, 3.1, and 4.1 in the _B. velezensis_ RVMD2 genome exhibit low similarity to known gene clusters, indicating a mixture of synthetic, regulatory, and related genes within these regions. Notably, Region 3.3 shows a 35% similarity to Bacillaene biosynthetic gene clusters, implying that it may produce compounds akin to Bacillaene. A 35% similarity suggests potential functional differences or the presence of novel derivatives. The other regions

**Table 6. Gene clusters involved in the synthesis of secondary metabolites from the *B. velezensis* RVMD2 genome as identified by the antiS-MASH software.**

| No | Region | Type | From | To | Most similar known cluster | Cluster Type | Similarity |
|----|--------|------|------|-----|---------------------------|--------------|------------|
| 1 | Region 1.1 | TransAT-PKS-like, NRPS | 9,282 | 65,583 | Bacillaene | Polyketide+NRP | 35% |
| 2 | Region 1.2 | NRPS, transAT-PKS, betalactone | 128,200 | 266,030 | Fengycin | NRP | 100% |
| 3 | Region 1.3 | Terpene | 288,597 | 310,480 | – | | – |
| 4 | Region 1.4 | T3PKS | 379,158 | 420,258 | – | | – |
| 5 | Region 1.5 | TransAT-PKS | 673,590 | 779,756 | Difficidin | Polyketide | 100% |
| 6 | Region 2.1 | other | 234,267 | 275,685 | Bacilysin | Other | 100% |
| 7 | Region 2.2 | RiPP-like, NRP-metallophore, NRPS | 831,891 | 883,683 | Bacillibactin | NRP | 100% |
| 8 | Region 3.1 | Terpene | 190,928 | 211,668 | – | | – |
| 9 | Region 3.2 | TransAT-PKS | 553,628 | 641,840 | Macrolactin H | Polyketide | 100% |
| 10 | Region 3.3 | TransAT-PKS, T3PKS, NRPS | 860,708 | 897,830 | Bacillaene | Polyketide+NRP | 71% |
| 11 | Region 4.1 | NRPS | 255,859 | 321,266 | Surfactin | NRP | 82% |
| 12 | Region 4.2 | NRPS, transAT-PKS, lanthipeptide-class-iii | 390,569 | 474,404 | Andalusicin A/andalusicin B | RiPP | 100% |

The 'From' and 'To' fields denote the locations of gene clusters that synthesize secondary metabolites in the *B. velezensis* RVMD2 genome. **NRPS** refers to Nonribosomal peptide synthetase, **PKS** to Polyketide synthetase, **RiPP** to Ribosomally synthesized and post-translationally modified peptide, and **T3PKS** to Type III polyketide synthases.

(1.3, 1.4, 3.1, and 4.1) do not match any known gene clusters. Based on the integrity and numerous functional genes in these clusters, it is likely that they represent new secondary metabolite synthesis gene clusters specific to *B. velezensis* RVMD2. These gene clusters highlight the diverse genetic functions and biosynthetic pathways present in the *B. velezensis* RVMD2 genome, showcasing its capability to produce a wide range of bioactive compounds.

## Comparative genome analysis and pangenome analysis

For Comparative genome analysis and pangenome analysis, we invested the huge capacity of IPGA to analysis large number of genomes at once, 614 out of 869 genomes of different *B. velezensis* strains available in the NCBI database as of June 26, 2024 were accepted by IPGA and phylogenetically analyzed, the phylogenic tree depend on the shared gene cluster were constructed. Fig 9 A presents a comprehensive pangenome analysis of 614 *B. velezensis* strains, including strain RVMD2. The rarefaction curves show that the pan-genome, represented by the blue curve, contains 22,752 gene clusters, reflecting a high level of genetic diversity within the species. In contrast, the core genome, represented by the orange curve, consists of 1,736 gene clusters, indicating a conserved set of genes shared across all strains. The continuous increase of the pan-genome with the addition of more genomes suggests that the size of the *B. velezensis* pan-genome may continue to grow and remains in an open state, indicating that the full genetic diversity of *B. velezensis* has not yet been fully captured. The early stabilization observed in the core genome suggests a consistent and conserved set of genes shared across all examined strains. This finding aligns with the results of a pangenome analysis conducted by Wang et al [66] on 46 *Bacillus velezensis* genomes. Building upon these findings, the 14 strains most closely related to RVMD2 were identified for further comparative analysis were: (GCA_000973585.1, GCA_001709115.1, GCA_001723375.1, GCA_002082365.1, GCA_004337655.1, GCA_006350975.1, GCA_013122275.1, GCA_014204475.1, GCA_017599365.1, GCA_018398955.1, GCA_018771665.1, GCA_023614465.1, GCA_904841115.1, and GCA_904842145.1). Further pan-genome profiling and COG annotation were performed on these strains, including *Bacillus velezensis* RVMD2, to categorize their genes. A heatmap and hierarchical clustering, based on pairwise average nucleotide identity (ANI) values among these closely related strains,

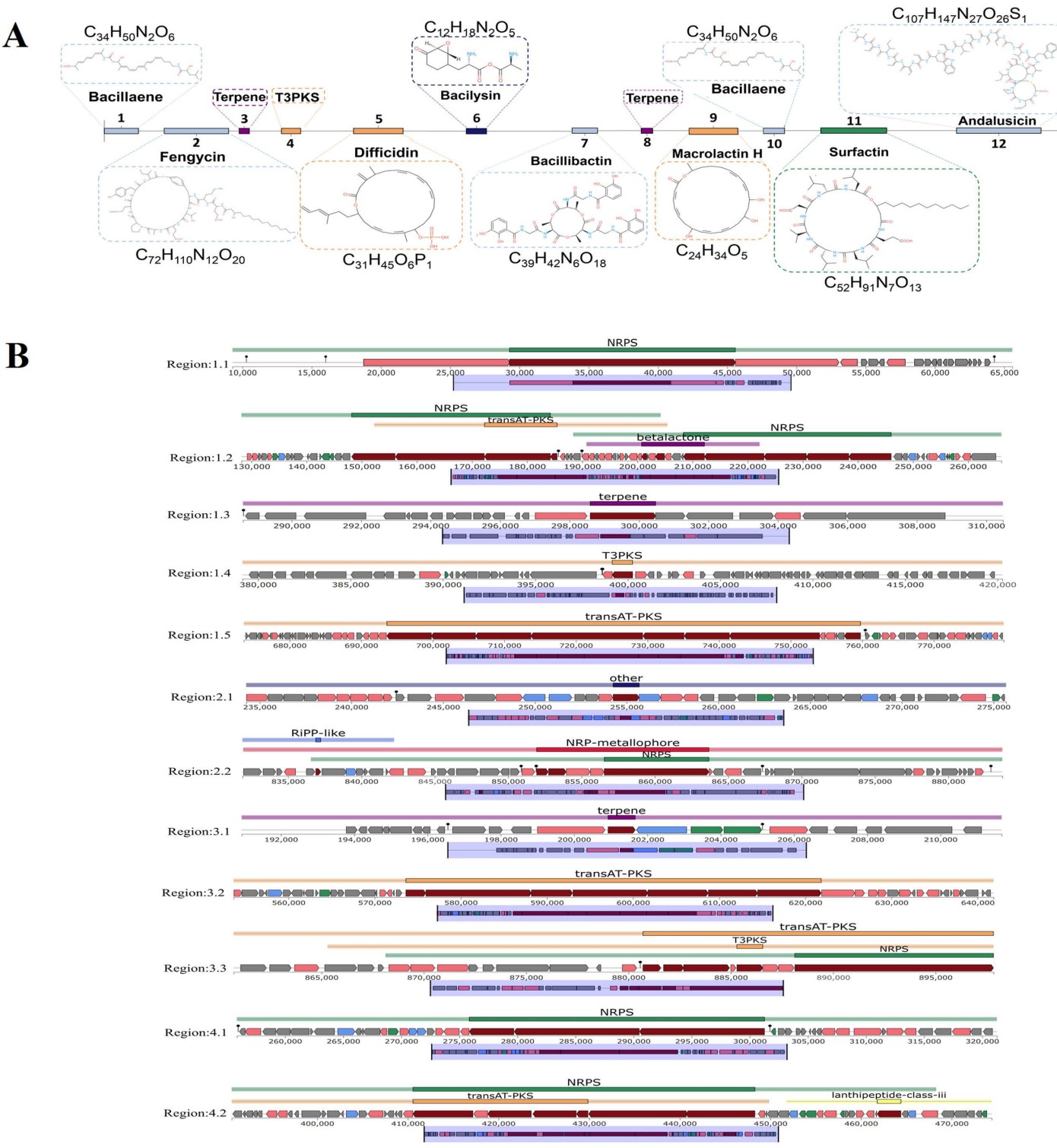

**Fig 8. Secondary metabolites and gene clusters in B. velezensis RVMD2 genome.** (A) Structures of secondary metabolites and gene clusters identified in the *B. velezensis* RVMD2 genome. (B) Schematic representation of gene clusters with diverse functions highlighted in different colors, as shown in the legend. Predictions were made using antiSMASH software version 4.0.

A

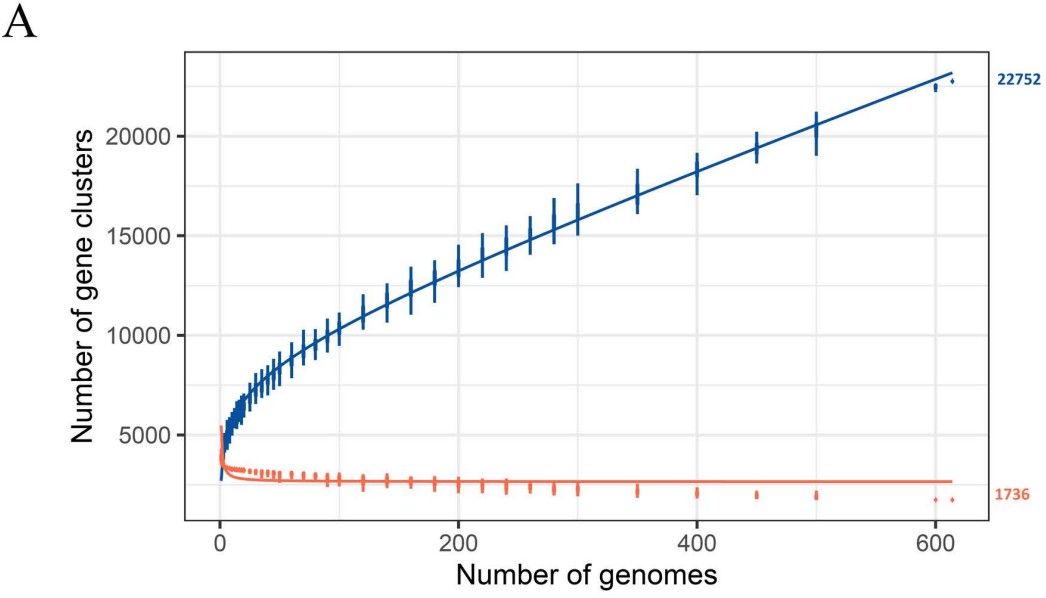

B

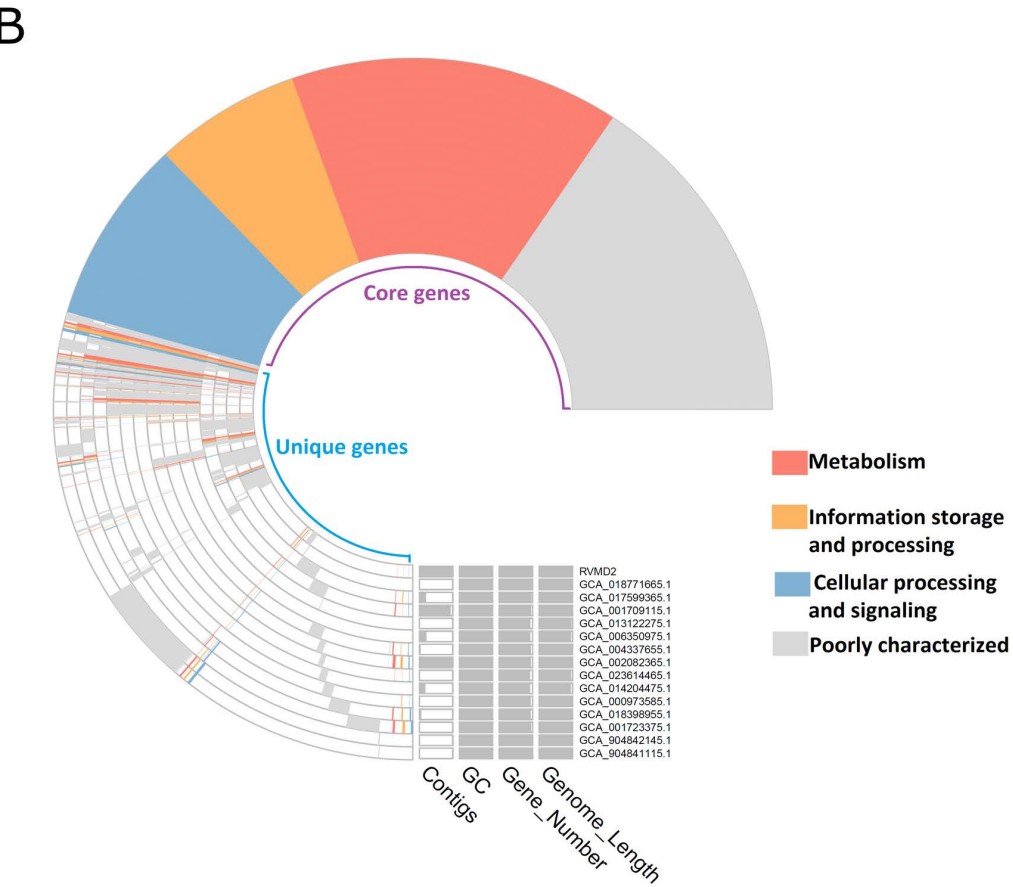

**Fig 9. Comprehensive pangenome analysis of *B. velezensis* strains, including detailed profile of strain RVMD2 and its closest 14 strains.** (A) Pangenome analysis of 615 *B. velezensis* strains, including strain RVMD2. The rarefaction curves compare pan-gene clusters (blue) and core gene clusters (orange), generated using IPGA. (B) Pangenome profile of *B. velezensis* strain RVMD2 and its closest 14 strains. COG annotation shows core and unique genes among these strains. The phylogenetic tree reflects the number of shared gene clusters.

is presented in S2 Fig. Additionally, Fig 9B offers detailed insights into the pangenome structure of *B. velezensis* RVMD2 and its 14 closest strains, revealing the classification of orthologous genes into core, accessory, and unique clusters. The core genome, composed of 3,440 genes, includes essential functions related to metabolism (highlighted in red), information storage (orange), and cellular signaling (blue). The pan-genome, with 5,667 genes, demonstrates overall genetic diversity, including 1,340 metabolic genes, 657 genes for information storage and processing, 818 genes for cellular processes and signaling, and 2,852 poorly characterized genes. RVMD2 has 45 unique gene clusters, suggesting distinct functional capabilities and potential adaptations. The majority of these unique genes are poorly characterized, opening the door for potential novel discoveries and products. The genomic data tables indicate variations in contig numbers (1–112), consistent GC content (~46%), gene numbers (3,830–4,147), and genome lengths (3,956,735–4,212,579 bp). This analysis emphasizes the stable core genome, significant genetic diversity from unique genes, and adaptive potential from accessory genes, providing a comprehensive understanding of the genetic makeup and evolutionary dynamics of *B. velezensis* strains.

Furthermore, the closest three strains (Q12, CFSAN034340, and ASM-2) were selected for further genome comparison (S3 Fig). Detailed visual whole-genome similarity analysis was performed using the FastANI 1.3.3 tool. Fig 10 shows a detailed visualization of similarity analysis between the assembled genome of *B. velezensis* strain RVMD2 and its three closest strains (Q12, CFSAN034340, and ASM-2) using the Proksee tool to determine the

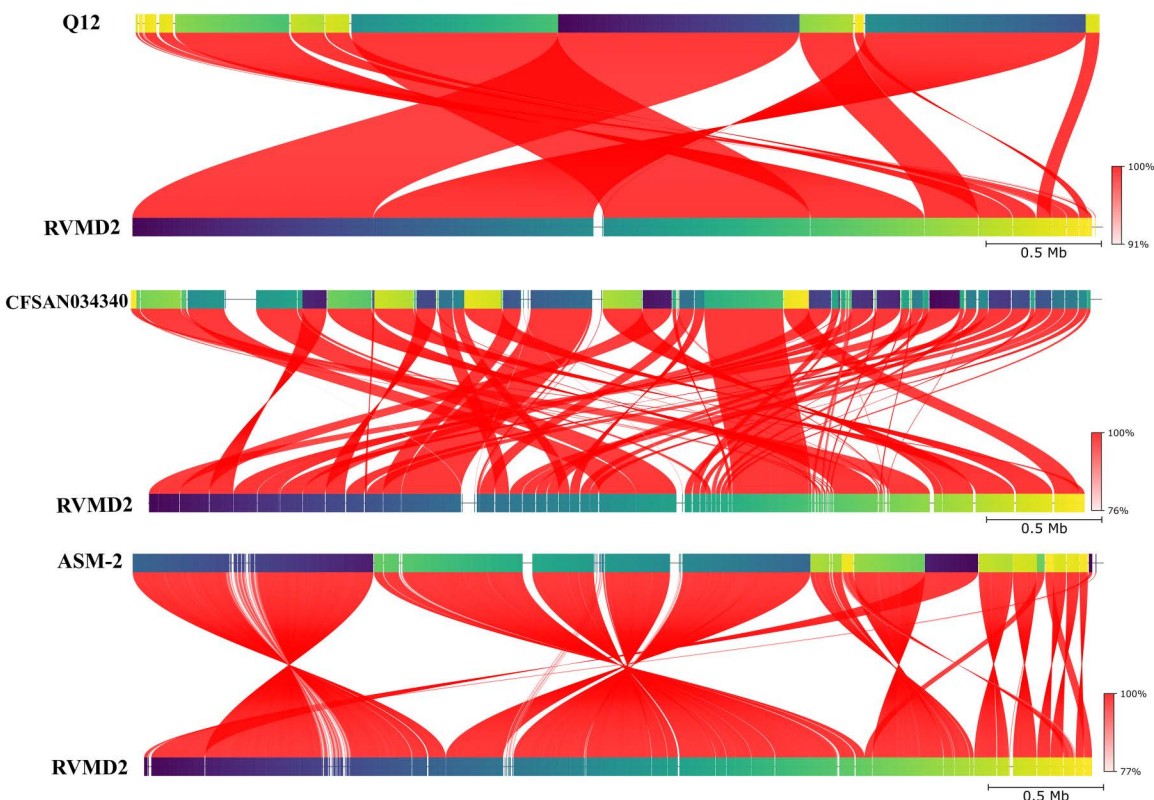

**Fig 10. Detailed visualization of similarity analysis between the assembled genome of *Bacillus velezensis* strain RVMD2 and its three closest *Bacillus velezensis* strains (Q12, CFSAN034340, and ASM-2), using the Proksee tool to determine the average nucleotide identity (FastANI).** Each red line represents a reciprocal map between the genomes, highlighting their evolutionarily conserved sequences. The red lines denote conserved genomic regions, with the color intensity indicating a high value of ANI. This whole genome alignment and comparison illustrate the genomic conservation and divergence between RVMD2 and its closest strains.

average nucleotide identity (FastANI). Each red line represents a reciprocal map between the genomes, highlighting their evolutionarily conserved sequences. The red lines denote conserved genomic regions, with the color intensity indicating a high value of ANI. The ANI values for Q12, CFSAN034340, and ASM-2 are 99.99%, 99.85%, and 99.18%, respectively. This whole genome alignment and comparison illustrate the high degree of genomic conservation and slight divergence between RVMD2 and its closest strains, emphasizing their evolutionary relationships and potential functional similarities.

Despite the promising findings, this study has several limitations. Firstly, while rigorous antiseptic measures were taken during sampling, the possibility remains that the isolate may have originated from external sources such as adjacent soil or air, rather than being definitively endogenous to the rock varnish layer. Secondly, functional validation of the identified genes and biosynthetic pathways through laboratory experiments was not conducted, which is essential to confirm their roles and potential applications. Thirdly, although genomic analysis suggests the presence of potential antibiotic resistance and virulence factors, phenotypic studies are needed to confirm their real-world implications for human health and disease. Additionally, the fragmented nature of the *Bacillus velezensis* RVMD2 genome assembly limits the ability to precisely determine structural rearrangements and genomic organization. Furthermore, this study did not investigate the ecological interactions or assess the environmental implications of utilizing *B. velezensis* RVMD2 in agricultural applications. Nevertheless, the analyses and tools employed align with standard practices and provide meaningful insights into the strain's genomic potential, laying the groundwork for future experimental and ecological studies

## Conclusion

The comprehensive integration of whole-genome sequencing, phylogenetic analysis, and functional annotation has unveiled significant insights into the genetic and functional profile of *B. velezensis* RVMD2. This strain, isolated from the harsh conditions of desert rock varnish in Ma'an, Jordan, exhibits promising potential as a source of bioactive compounds and traits that promote plant growth. Its high genomic similarity to other *Bacillus* species, along with the identification of several secondary metabolite biosynthesis gene clusters, positions *B. velezensis* RVMD2 as a strong candidate for use in agricultural applications, biocontrol, and bioremediation. Notably, the genome harbors genes associated with the production of antimicrobial compounds such as Bacillaene, Fengycin, Difficidin, and Surfactin. Additionally, the discovery of novel biosynthetic gene clusters indicates the potential for the production of previously unknown bioactive compounds. Notably, the genome harbors genes associated with the production of antimicrobial compounds such as Bacillaene, Fengycin, Difficidin, and Surfactin. Additionally, the discovery of novel biosynthetic gene clusters suggests the potential for producing previously unknown antimicrobial compounds, thereby contributing to the growing interest in novel drug discovery. The strain's genetic repertoire for plant growth promotion, heavy metal resistance, and bioremediation underscores its relevance in addressing agricultural challenges, such as improving crop resistance and reducing reliance on chemical pesticides. Functional annotation highlights the presence of genes involved in various metabolic processes, stress responses, and virulence factors, emphasizing the strain's biological complexity and its industrial relevance. The open pan-genome and conserved core genome of B. *velezensis* RVMD2 reflect its evolutionary adaptability and its promise as a resource for enhancing crop resilience and developing eco-friendly biopesticides. Pan-genome analysis reveals substantial genetic diversity and adaptability among *B. velezensis* strains, with an open pan-genome and a well-conserved core genome. This genetic diversity, coupled with the evolutionary resilience of the strain, suggests that *B. velezensis* RVMD2 and its related strains are capable of thriving in diverse environments, positioning them as valuable subjects for future research and biotechnological development. Future research should focus on functional validation of the identified biosynthetic pathways and their potential applications in real-world agricultural settings, paving the way for new innovations in sustainable farming and biotechnological advancements.

 

## Supporting information

**S1 Table. Taxonomic classification and significance levels for the query dataset, analyzed through MiGA (Microbial Genome Atlas.** The classification is inferred by the maximum Average Amino Acid Identity (AAI) found against all genomes in the database. The *p*-value, estimated from the empirical distribution observed in all NCBI RefSeq reference genomes, indicates the probability of a different classification with the observed AAI. Closest relatives identified were *Bacillus velezensis* NZ CP036527 (99.86% ANI) and *Bacillus velezensis* NZ CP010556 (99.35% ANI). Significance at *p*-value below: ***0.05, ****0.01.
(DOCX)

**S2 Table. Pairwise dDDH values between *Bacillus velezensis* strain RVMD2 genome and Type-Strain Genomes.** Displays the pairwise digital DNA-DNA hybridization (dDDH) values, including confidence intervals (C.I.), between RVMD2 genome and selected type-strain genomes using GBDP formula *d4* (GGDC formula 2), which sums all identities found in high-scoring pairs (HSPs) divided by the overall HSP length.
(DOCX)

**S3 Table. Distribution of Carbohydrate-Active Enzymes (CAZys) in the *Bacillus velezensis* RVMD2 genome as analyzed by Protologger v0.99.**
(DOCX)

**S4 Table. Prophage regions identified in *Bacillus velezensis* strain RVMD2 genome using PHASTEST [34] Includes region length, completeness, score, total proteins, position, most common phage, and GC content.**
(DOCX)

**S1 Fig. Synteny and Phylogenetic Comparison of Genomic Island Content.** Genomic islands, representing probable horizontally acquired gene clusters, were identified and compared with close phylogenetic neighbors using IslandCompare (v1.0) https://islandcompare.ca/. This analysis distinguishes strain RVMD2 from its neighbors based on the number, type, and position of these islands. The left panel shows the phylogenetic tree, while the right panel displays synteny blocks and unique genomic islands, highlighting strain RVMD2.
(TIF)

**S2 Fig. Heatmap and hierarchical clustering based on pairwise average nucleotide identity (ANI) values of B. velezensis strain RVMD2 and its 14 closest strains.** The analysis was performed using The Integrated Pan-Genome Analyser (IPGA) (https://nmdc.cn/ipga/), a web-based service. The heatmap illustrates the ANI values, with the color gradient representing the percentage identity, ranging from 95% to 100%. The dendrogram on the right shows the hierarchical clustering, indicating the evolutionary relationships among the strains.
(TIF)

**S3 Fig. The Venn diagram, constructed using the OrthoVenn3 online service, displays the distribution of shared gene families (orthologous clusters) among the B. velezensis strain RVMD2 and its three closest Bacillus velezensis strains (ASM-2, Q12, and CFSAN034340).** The overlapping regions indicate the number of gene families shared between the strains, while unique regions indicate gene families specific to each strain. The bar chart below the Venn diagram represents the size of each gene list for the strains. The bottom histogram shows the number of elements specific to one list or shared by two, three, or all four lists.
(TIF)

## Author contributions

**Conceptualization:** Sulaiman M Alnaimat, Saqr Abushattal, Saif M Dmour.

---

**Data curation:** Sulaiman M Alnaimat, Saqr Abushattal, Saif M Dmour.

**Formal analysis:** Sulaiman M Alnaimat, Saqr Abushattal.

**Investigation:** Saif M Dmour.

**Methodology:** Sulaiman M Alnaimat, Saqr Abushattal, Saif M Dmour, Wajdy J. Al-Awaida, Amani M Ayyash.

**Project administration:** Saqr Abushattal.

**Resources:** Saqr Abushattal.

**Software:** Sulaiman M Alnaimat, Saqr Abushattal, Wajdy J. Al-Awaida, Khang Wen Goh.

**Supervision:** Sulaiman M Alnaimat, Saif M Dmour.

**Validation:** Sulaiman M Alnaimat, Saqr Abushattal, Saif M Dmour, Khang Wen Goh.

**Visualization:** Saif M Dmour.

**Writing – original draft:** Sulaiman M Alnaimat, Saqr Abushattal, Saif M Dmour, Wajdy J. Al-Awaida, Amani M Ayyash, Khang Wen Goh.

**Writing – review & editing:** Sulaiman M Alnaimat, Saqr Abushattal, Saif M Dmour, Wajdy J. Al-Awaida, Amani M Ayyash, Khang Wen Goh.

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
