## [Decision Letter · Decision Letter 0]

27 Dec 2024

PONE-D-24-46090Whole-Genome Sequencing and Comparative Genomics Reveal Prolific Bioactive Traits in Bacillus velezensis RVMD2 from Desert Rock Varnish in Ma'an, JordanPLOS ONE

Dear Dr. Al-Awaida,

Thank you for submitting your manuscript to PLOS ONE. After careful consideration, we feel that it has merit but does not fully meet PLOS ONE’s publication criteria as it currently stands. Therefore, we invite you to submit a revised version of the manuscript that addresses the points raised during the review process.

Dear Prof.,Kindly revise your manuscript according to the reviewers' comments. 

We look forward to receiving your revised manuscript.

Kind regards,

Kamal Ahmad Qureshi, PhD

Academic Editor

PLOS ONE

Journal Requirements:

[This research was funded by the Deanship of Scientific Research at Al-Hussein Bin Talal University, Jordan, under grant number 2022/119].

Reviewers' comments:

Reviewer's Responses to Questions

**Comments to the Author**

1. Is the manuscript technically sound, and do the data support the conclusions?

Reviewer #1: Yes

Reviewer #2: Yes

Reviewer #3: Yes

Reviewer #4: Yes

2. Has the statistical analysis been performed appropriately and rigorously? 

Reviewer #1: I Don't Know

Reviewer #2: Yes

Reviewer #3: N/A

Reviewer #4: Yes

3. Have the authors made all data underlying the findings in their manuscript fully available?

Reviewer #1: Yes

Reviewer #2: Yes

Reviewer #3: Yes

Reviewer #4: Yes

4. Is the manuscript presented in an intelligible fashion and written in standard English?

Reviewer #1: Yes

Reviewer #2: Yes

Reviewer #3: Yes

Reviewer #4: Yes

5. Review Comments to the Author

Reviewer #1: a) Problem Statement

The manuscript lacks a clear and compelling problem statement. While it emphasizes the extremophilic origin of Bacillus velezensis RVMD2, it does not adequately frame the scientific gap or justify the novelty of isolating this strain.

Can you introduce a specific research problem, such as the underexplored bioactive potential of microorganisms from desert varnish, and align it with broader biotechnological applications.

b) Objectives

The study outlines various analyses (e.g., genomic, phylogenetic, bioactive compound identification) but does not explicitly state clear, measurable objectives.

Please clarify which to achieve?

To determine the genomic adaptations of B. velezensis RVMD2 to extreme environments or ANTIMICROBIAL or BIOREMEDIATION?

c) Methods and Environmental Relevance

There is an inconsistency in claiming the extremophile nature of the strain versus its cultivation at 37°C in a controlled laboratory setting. This raises concerns about how well the findings translate to its natural habitat.

d) Strength of Claims

The manuscript suggests the presence of novel antimicrobial compounds and bioactive traits BUT lacks experimental validation or biochemical characterization to substantiate these claims, can you couple your claim with prooof on antimicrobial potential through wet-lab experiments.

e) Genomic and Comparative Analysis: The comparative analysis is detailed but focuses heavily on the technical aspects without adequately linking findings to practical implications.

f) The manuscript provides a robust genomic dataset but falls short in demonstrating its relevance and potential applications.

Reviewer #2: The manuscript entitled "Whole-Genome Sequencing and Comparative Genomics Reveal Prolific Bioactive Traits in Bacillus velezensis RVMD2 from Desert Rock Varnish in Ma'an, Jordan" is well written and reports significant results. I have gone through the manuscript, its almost good, however there are some minor grammatical typographical errors which needs to be updated, such as;

1. Line 20: Italicize, B. velezensis

2. Line 22: Italicize the genus name in B. velezensis

3. Line 39: "rock varnish—a thin", check and remove extra space

4. Line 40-42: Some more recent literature on the whole genome sequencing and genome mining of microbes from extreme environments such as deserts can be cited e.g Fatima et al., 2024, https://doi.org/10.1016/j.genrep.2024.102050

5. Line 46: Italicize "Bacillus" this a frequent error, check the manuscript thoroughly for scientific names and italicize

6. Lines 57-61: why all these lines are italic, check and upadte

7. Line 77: Italicize "Botryosphaeria dothidea"

8. Line 98: Check coordinates, there must be degree sign on the coordinates

9. Fig quality is almost fine however the figure of circular genome and others figure such as Fig 1, Fig 5, Fig 7, Fig 8 can be improved further for clarity

10. The methodology showed the WGS data was also analyzed on AntiSmash however the AntiSmash output table is not visible in the results, although the Table for BV-BRC analysis is given which shows only the heavy metal resistant and plant growth promoting gene clusters, check and add antiSmash output Table in the results

Reviewer #3: The research article by Alnaimat et al. aimed determine the taxonomic identity and provide a comprehensive characterization of the phylogenetic, genomic, and taxonomic features of Bacillus velezensis strain RVMD2.To this purpose the authors establish its taxonomic placement through phylogenomic analysis and examine its genes related to antibiosis and other distinctive traits, comparing them with those from the broader Bacillus genus. Using a multiphase classification strategy that integrates whole-genome shotgun sequencing with rRNA gene amplicon analysis. Overall, the paper was well written, and the data look quite convincing. However, some points need clarification.

Comments to the Authors

The genome was sequenced resulting in a 4,212,579 but the final structure of the chromosome was not obtained since, the assembled genome comprises 112 contigs. It is because of this that the localization of different elements in the chromosome or the comparisons with other strains chromosomes made no sense. For example, figure 2, presented the chromosome in close circular form; figure 7 presented different genomics elements in specific localizations on the circular chromosome, and in figure 10 the comparison with other chromosomes (the observed inversions may be artifacts).

Reviewer #4: The manuscript by Alnaimat et al explains their work characterizing a Bacillus strains sample at the Ma'an desert. Overall I think the quality of the work and the writing is very good, with solid evidence supporting their claims. I have some comments for the authors to consider, I hope they are useful and can improve the article:

L193 I assume these datasets are currently under embargo, please confirm.

L239 What are the distance units? Please indicate.

L375 "ANI coverage of 91.1648%, which is higher than the ANI threshold cut-off value (95–96%)". Please rephrase, currently it is confusing as 91 < 95.

L401 "tree places B. velezensis strain RVMD2 in a well-supported clade with high bootstrap values, confirming its close relationship with other B. velezensis strains". I guess the fact that the closest clade contains 3 different species suggests that more species/strains should be included to increase the resolution, don't you agree? This links to my next comment.

L408 The legend or the text should explain how Bacillus species included were selected.

L518 In the version available for review it's impossible to read anything from this figure, please make sure the resolution is good enough.

L635 I wonder how much sense this makes considering the number of contigs in the assembly

Minor issues:

L57 please remove italics after species name

L139 replace . with ,

L154 add a space before web please

L395 "we designating it" please change to "designated"

L719 Please add journal name and complete citation if needed

6. PLOS authors have the option to publish the peer review history of their article (what does this mean? ). If published, this will include your full peer review and any attached files.

**Do you want your identity to be public for this peer review?** For information about this choice, including consent withdrawal, please see our Privacy Policy .

Reviewer #1: **Yes: ** Fahrul Huyop

Reviewer #2: No

Reviewer #3: No

Reviewer #4: No

---

## [Author Response · Author response to Decision Letter 1]

7 Jan 2025

Dr. Kamal Ahmad Qureshi, PhD

Academic Editor

PLOS ONE

Subject: Response to Reviewer Comments for Manuscript PONE-D-24-46090

Dear Dr. Qureshi,

We thank you and the reviewers for the thorough evaluation of our manuscript titled "Whole-Genome Sequencing and Comparative Genomics Reveal Prolific Bioactive Traits in Bacillus velezensis RVMD2 from Desert Rock Varnish in Ma'an, Jordan" (Manuscript ID: PONE-D-24-46090). We have carefully addressed all the reviewers’ comments and the journal's requirements, and we have revised the manuscript accordingly.

In summary, we have made the following key updates:

1. Journal Requirements:

o Included a statement in the Methods section clarifying that no permits were required for the fieldwork, as the site is publicly accessible, and no protected or endangered species were involved.

o Revised the funding disclosure to clarify the funders’ role, stating explicitly: "The funders had no role in study design, data collection and analysis, decision to publish, or preparation of the manuscript."

o We confirm that all data associated with this manuscript are now freely available without any restrictions. The data have been deposited in publicly accessible repositories, and the relevant accession numbers or identifiers (IDs) are provided within the manuscript to ensure full transparency and accessibility.

2. Reviewer Comments:

o Addressed all major and minor reviewer comments in detail, including corrections to technical inaccuracies and enhancements to figure captions.

o Clarified methodological details, such as phylogenetic analyses, ANI interpretations, and figure resolutions, to improve transparency and readability.

We have attached a detailed point-by-point response to all comments, highlighting how each concern was addressed in the revised manuscript.

We greatly appreciate the reviewers’ constructive feedback, which has helped us improve the clarity and rigor of our study. We are confident that the revised manuscript now meets the standards of PLOS ONE and respectfully submit it for further consideration.

Thank you for your time and attention. We look forward to your feedback.

Sincerely,

Dr. Al-Awaida

Reviewer #1:

a) Problem Statement: The manuscript lacks a clear and compelling problem statement. While it emphasizes the extremophilic origin of Bacillus velezensis RVMD2, it does not adequately frame the scientific gap or justify the novelty of isolating this strain. Can you introduce a specific research problem, such as the underexplored bioactive potential of microorganisms from desert varnish, and align it with broader biotechnological applications viewer?

b) Objectives: The study outlines various analyses (e.g., genomic, phylogenetic, bioactive compound identification) but does not explicitly state clear, measurable objectives. Please clarify which to achieve?

To determine the genomic adaptations of B. velezensis RVMD2 to extreme environments or ANTIMICROBIAL or BIOREMEDIATION?

Our response: To address these concerns, we have revised the manuscript to provide a clear research gap and align the study with broader biotechnological applications. Additionally, we have explicitly stated the objectives to ensure they are measurable and directly aligned with the study's goals. In the Revised Manuscript with Track Changes (Line 92), we have included the following updated text:

• To the best of our knowledge, no studies have comprehensively investigated the bioactive gene potential of microorganisms thriving in this unique environment. Desert varnish, an underexplored habitat, may harbor microbial communities with novel genes and secondary metabolites that could have significant applications in medicine and agriculture. Recognizing the need to explore this potential, this study focuses on Bacillus velezensis strain RVMD2, isolated from rock varnish in the Ma'an Desert, Jordan.

• This study aims to determine the taxonomic identity and provide a comprehensive characterization of the phylogenetic, genomic, and taxonomic features of B. velezensis RVMD2. Specifically, we seek to establish its taxonomic placement through phylogenomic analysis and investigate genes related to antibiosis, secondary metabolite biosynthesis, and other distinctive traits. These features are compared with those from the broader Bacillus genus to assess its potential contributions to biotechnological applications. To achieve these goals, we employ a multiphase classification strategy that integrates whole-genome shotgun sequencing with rRNA gene amplicon analysis, providing a robust framework for understanding the strain's genomic potential.

c) Methods and Environmental Relevance: There is an inconsistency in claiming the extremophile nature of the strain versus its cultivation at 37°C in a controlled laboratory setting. This raises concerns about how well the findings translate to its natural habitat.

Our response: We would like to clarify that we do not claim Bacillus velezensis RVMD2 itself is an extremophile. Instead, we emphasize that it was isolated from an extreme environment—desert rock varnish—which is characterized by harsh conditions such as high temperatures, UV exposure, and limited nutrients. The occurrence of non-extremophilic microorganisms in extreme environments is well-documented, as such organisms often possess specific adaptive features that enable them to survive under these challenging conditions. We believe that B. velezensis RVMD2 may exhibit similar adaptations, and our study aims to explore its genomic features that could contribute to its survival in such an environment.

d) Strength of Claims: The manuscript suggests the presence of novel antimicrobial compounds and bioactive traits BUT lacks experimental validation or biochemical characterization to substantiate these claims, can you couple your claim with proof on antimicrobial potential through wet-lab experiments.

Our response: As stated in the limitations section of our manuscript, the bioactive potential identified in the genome requires further experimental validation in the laboratory. This will be the focus of our upcoming project, where we plan to test B. velezensis RVMD2 for its bioactive metabolites and employ techniques such as HPLC, mass spectrometry, and antimicrobial assays to fully characterize these metabolites and confirm their novelty. We believe that the work presented here provides a strong foundation and represents a crucial first step in understanding the genomic potential of this strain. By identifying biosynthetic gene clusters (BGCs) and highlighting the strain’s biosynthetic capabilities, this study sets the stage for future investigations into its biochemical properties and practical applications.

e) Genomic and Comparative Analysis: The comparative analysis is detailed but focuses heavily on the technical aspects without adequately linking findings to practical implications.

f) The manuscript provides a robust genomic dataset but falls short in demonstrating its relevance and potential applications.

Our response: As highlighted before, the current study primarily focuses on characterizing the genomic features of Bacillus velezensis RVMD2, including its biosynthetic gene clusters (BGCs) and adaptive traits. While we have provided a robust genomic dataset, we acknowledge the need to more explicitly highlight its relevance and potential applications. This will be a central focus of our upcoming project, where we plan to test B. velezensis RVMD2 for its bioactive metabolites and employ techniques such as HPLC, mass spectrometry, and antimicrobial assays to fully characterize these metabolites and confirm their novelty. These experiments will allow us to explore the strain’s potential applications in medicine, agriculture, and other biotechnological fields.

Reviewer #2:

• Line 20: Comment: Italicize B. velezensis.

Our response: This correction has been implemented in the manuscript.

• Line 22:Comment: Italicize the genus name in B. velezensis.

Our response: The genus name has been italicized as requested throughout the manuscript.

• Line 39: Comment: "rock varnish—a thin" has an extra space; remove it.

Our response: The extra space has been removed for consistency and readability.

• Line 40–42: Our response: Some more recent literature on whole-genome sequencing and genome mining of microbes from extreme environments, such as deserts, should be cited. For example: Fatima et al., 2024, https://doi.org/10.1016/j.genrep.2024.102050.

Our response: Thank you for the suggestion. The following text has been added to Line 44 in the revised manuscript: "Genome mining of desert-isolated strains has enabled the identification of biosynthetic gene clusters (BGCs) linked to the production of diverse bioactive compounds, including antibiotics (4). These findings underscore the potential of extreme environment-derived microorganisms as promising sources for novel antibiotic discovery."

• Line 46: Comment: Italicize Bacillus. This error occurs frequently in the manuscript; check and correct all scientific names.

Our response: All scientific names, including Bacillus, have been reviewed and italicized consistently throughout the manuscript.

• Lines 57–61: Comment: Why are all these lines italicized? Check and update.

Our response: The formatting issue has been reviewed, and unnecessary italicization has been corrected.

• Line 77: Comment: Italicize Botryosphaeria dothidea.

Our response: This correction has been implemented.

• Line 98: Comment: Check the coordinates; ensure the degree sign (°) is present.

Our response: The coordinates have been reviewed, and the degree sign (°) has been added where necessary.

• Figures: Comment: The quality of most figures is acceptable, but the resolution of the circular genome and figures such as Fig. 1, Fig. 5, Fig. 7, and Fig. 8 should be improved for clarity.

Our response: The resolution of the mentioned figures has been improved to ensure clarity and compliance with publication standards.

• AntiSMASH Output: Comment: The methodology mentions analyzing WGS data with AntiSMASH, but the AntiSMASH output table is missing from the results. Include the table in the results section.

Our response: We thank the reviewer for their observation. The table detailing gene clusters involved in secondary metabolite synthesis, previously included as Table S5 in the supplementary files, has now been moved to the main text of the manuscript for better visibility, please check Table 6. This table complements the visualization of secondary metabolite gene clusters presented in Figure 8. We hope this adjustment addresses the reviewer’s concern and enhances the clarity of our results.

Reviewer #3:

• The genome was sequenced, resulting in 4,212,579 bp, but the final structure of the chromosome was not obtained. The assembly comprises 112 contigs, making chromosomal localization or comparisons with other chromosomes problematic. For example, the observed inversions may be artifacts.

Our response: We thank the reviewer for their insightful comments regarding the genome assembly and its implications for chromosomal localization and structural comparisons. We acknowledge that the Bacillus velezensis RVMD2 genome assembly consists of 112 contigs, which limits the precise identification of structural rearrangements and genomic organization. However, we would like to clarify that the tools used in this study—such as Proksee, IslandViewer 4, PHASTEST, and FastANI—are specifically designed to handle fragmented genomes and remain reliable for visualizing genomic features and calculating average nucleotide identity (ANI).

For example:

• Figure 2 (circular genome map) illustrates genomic features such as GC content, GC skew, and open reading frames (ORFs), providing an overview of the strain’s genomic structure.

• Figure 7 highlights genomic islands and prophage regions predicted using IslandViewer 4 and PHASTEST, respectively. These predictions are based on sequence composition and annotations and are considered reliable despite the fragmented assembly.

• Figure 10 demonstrates conserved genomic regions and alignments with closely related strains, supported by high ANI values (99.99%, 99.85%, and 99.18% with strains Q12, CFSAN034340, and ASM-2, respectively), which validate the evolutionary relationships and functional similarities.

While the fragmented nature of the assembly may introduce some limitations, such as potential artifacts in structural comparisons (e.g., inversions), the results align with standard practices in bacterial genomics and provide meaningful insights into the genomic potential of B. velezensis RVMD2.

To address these points, we have updated the manuscript as follows:

1. Added clarification in the results section (Lines 518-522):

"The genomic islands were identified using IslandViewer 4 [32], and the prophage regions were predicted using PHASTEST [34]. These predictions are based on sequence composition and annotations, and while the level of fragmentation in the genome assembly may have some influence, the results are still considered reliable and align with standard practices in bacterial genomics."

2. Updated the limitations section (Lines 688-702)::

"The fragmented nature of the Bacillus velezensis RVMD2 genome assembly also limits the ability to precisely determine structural rearrangements and genomic organization. Nevertheless, the analyses and tools employed align with standard practices and provide meaningful insights into the strain’s genomic potential."

Reviewer #4

• L193: Confirm whether datasets are currently under embargo.

Our response: All data mentioned in this manuscript, including the Whole Genome Shotgun (WGS) project, raw FASTQ files from genome sequencing, and Sequence Read Archive (SRA) data, are now publicly available without any restrictions. The relevant accession numbers or identifiers (IDs) are provided within the manuscript to ensure full transparency and accessibility.

• L239: What are the distance units? Please indicate.

Our response: The scale bar represents a genetic distance of 0.01 substitutions per site. The figure caption has been updated in the revised manuscript to include this information for clarity.

• L375: "ANI coverage of 91.1648%, which is higher than the ANI threshold cut-off value (95–96%)". Rephrase this statement for clarity, as 91 < 95.

Our response: Thank you for pointing out this inconsistency. Upon review, we realized that the text mistakenly referred to the ANI coverage (91.1648%) instead of the ANI percentage (97.803%), which is indeed above the species threshold cut-off value of 95–96%. The corrected text now reads:

"Based on Table 4, the analysis using the EzBiome Genome-ID tool identified Bacillus velezensis as the top match for bacterial isolate RVMD2, with an ANI of 97.803%, 16S similarity of 99.92%, recA identity of 98.563%, rplC identity of 99.524%, Mash identity of 97.8%, and ANI coverage of 91.1648%. The ANI percentage (97.803%) is higher than the species demarcation threshold of 95–96% (44)."

• L401: The phylogenetic tree places B. velezensis strain RVMD2 in a well-supported clade with high bootstrap values. However, the closest clade contains three different species, suggesting that more species/strains should be included to improve resolution. Agree?

Our response: The phylogenetic tree was constructed using the TYGS (Type Strain Genome Server) platform, which automatically selects related species based on the query genome (B. velezensis RVMD2). We did not manually curate the species included in the analysis, as TYGS determines the most relevant genomes for comparison.

All genomes within the clade containing our bacterial strain are part of the same species cluster. This aligns with the reclassification of B. methylotrophicus KACC 13105 and B. amyloliquefaciens subsp. plantarum FZB42 as synonyms of B. velezensis NRRL B-41580, as established by Dunlap et al. (2016). Consequently, we designate our strain as B. velezensis RVMD2, consistent with this taxonomic framework.

• L408: Explain how the Bacillus species included in the study were selected.

Our response: The Bacillus species included in the study were determined using the TYGS platform, which identifies the most relevant related

---

## [Decision Letter · Decision Letter 1]

21 Jan 2025

PONE-D-24-46090R1Whole-Genome Sequencing and Comparative Genomics Reveal Prolific Bioactive Traits in Bacillus velezensis RVMD2 from Desert Rock Varnish in Ma'an, JordanPLOS ONE

Dear Dr. Al-Awaida,

Thank you for submitting your manuscript to PLOS ONE. After careful consideration, we feel that it has merit but does not fully meet PLOS ONE’s publication criteria as it currently stands. Therefore, we invite you to submit a revised version of the manuscript that addresses the points raised during the review process.

Dear Author,

Thank you for addressing many of the comments. However, it appears that the comments from Reviewer 1 have not been fully or adequately addressed. Kindly review those comments carefully and make the necessary revisions to address them appropriately.

We look forward to receiving your revised manuscript.

Kind regards,

Kamal Ahmad Qureshi, PhD

Academic Editor

PLOS ONE

Journal Requirements:

Reviewers' comments:

Reviewer's Responses to Questions

**Comments to the Author**

1. If the authors have adequately addressed your comments raised in a previous round of review and you feel that this manuscript is now acceptable for publication, you may indicate that here to bypass the “Comments to the Author” section, enter your conflict of interest statement in the “Confidential to Editor” section, and submit your "Accept" recommendation.

Reviewer #1: (No Response)

Reviewer #2: All comments have been addressed

Reviewer #3: All comments have been addressed

Reviewer #4: All comments have been addressed

2. Is the manuscript technically sound, and do the data support the conclusions?

Reviewer #1: Yes

Reviewer #2: Yes

Reviewer #3: Yes

Reviewer #4: Yes

3. Has the statistical analysis been performed appropriately and rigorously? 

Reviewer #1: N/A

Reviewer #2: N/A

Reviewer #3: N/A

Reviewer #4: Yes

4. Have the authors made all data underlying the findings in their manuscript fully available?

Reviewer #1: Yes

Reviewer #2: Yes

Reviewer #3: Yes

Reviewer #4: Yes

5. Is the manuscript presented in an intelligible fashion and written in standard English?

Reviewer #1: No

Reviewer #2: Yes

Reviewer #3: Yes

Reviewer #4: Yes

6. Review Comments to the Author

Reviewer #1: The most important comments did not properly addressed.

The title still not match to the objectives of studies. This is fundamental and not properly addressed.

Reviewer #2: (No Response)

Reviewer #3: On reading through the "new version” of the manuscript, I was made aware of the many changes that have been made to fully explain all points. All suggestions and concerns reported in my previous evaluation report have been addressed carefully.

The authors have satisfied all of my concerns.

Reviewer #4: Thanks for replying to my comments and requests, they have all been address. My only remaining concern is the quality/resolution of the figures, but I assume the real ones, not the ones in my merged PDF, look ok. I leave that to the editor

7. PLOS authors have the option to publish the peer review history of their article (what does this mean? ). If published, this will include your full peer review and any attached files.

**Do you want your identity to be public for this peer review?** For information about this choice, including consent withdrawal, please see our Privacy Policy .

Reviewer #1: No

Reviewer #2: No

Reviewer #3: No

Reviewer #4: No

---

## [Author Response · Author response to Decision Letter 2]

28 Jan 2025

Response to Reviewer #1:

Thank you for your feedback. We have carefully revised the title to ensure it accurately reflects the study’s objectives. The new title, "Genomic insights into the taxonomic status and bioactive gene cluster profiling of Bacillus velezensis RVMD2 isolated from desert rock varnish in Ma'an, Jordan" explicitly highlights the key research components, including taxonomic classification, genomic characterization, and secondary metabolite gene analysis. We believe this revision aligns well with the study’s scope and objectives

---

## [Editor Report · Decision Letter 2]

31 Jan 2025

Genomic insights into the taxonomic status and bioactive gene cluster profiling of Bacillus velezensis RVMD2 isolated from desert rock varnish in Ma'an, Jordan

PONE-D-24-46090R2

Dear Dr. %Wajdy Jum’ah Al-Awaida%,

We’re pleased to inform you that your manuscript has been judged scientifically suitable for publication and will be formally accepted for publication once it meets all outstanding technical requirements.

Kind regards,

Kamal Ahmad Qureshi, PhD

Academic Editor

PLOS ONE
---

## [Editor Report · Acceptance letter]

PONE-D-24-46090R2

PLOS ONE

Dear Dr. Al-Awaida,

I'm pleased to inform you that your manuscript has been deemed suitable for publication in PLOS ONE. Congratulations! Your manuscript is now being handed over to our production team.

Kind regards,

on behalf of

Dr. Kamal Ahmad Qureshi

Academic Editor

PLOS ONE